# Enhanced antitumoral activity of TLR7 agonists via activation of human endogenous retroviruses by HDAC inhibitors

David Díaz-Carballo[1✉], Sahitya Saka [1], Ali H. Acikelli [1], Ekaterina Homp[1], Julia Erwes[1], Rebecca Demmig[1], Jacqueline Klein [1], Katrin Schröer [1], Sascha Malak[1], Flevy D'Souza[1], Adrien Noa-Bolaño [1], Saskia Menze[1], Emilio Pano [1], Swetlana Andrioff [1], Marc Teipel[1], Philip Dammann[2], Diana Klein [3], Amber Nasreen[4], Andrea Tannapfel[5], Nicole Grandi[6], Enzo Tramontano[6], Crista Ochsenfarth [7] & Dirk Strumberg[1]

In this work, we are reporting that "Shock and Kill", a therapeutic approach designed to eliminate latent HIV from cell reservoirs, is extrapolatable to cancer therapy. This is based on the observation that malignant cells express a spectrum of human endogenous retroviral elements (HERVs) which can be transcriptionally boosted by HDAC inhibitors. The endoretroviral gene *HERV-V2* codes for an envelope protein, which resembles syncytins. It is significantly overexpressed upon exposure to HDAC inhibitors and can be effectively targeted by simultaneous application of TLR7/8 agonists, triggering intrinsic apoptosis. We demonstrated that this synergistic cytotoxic effect was accompanied by the functional disruption of the TLR7/8-NFκB, Akt/PKB, and Ras-MEK-ERK signalling pathways. CRISPR/Cas9 ablation of *TLR7* and *HERV-V1/V2* curtailed apoptosis significantly, proving the pivotal role of these elements in driving cell death. The effectiveness of this new approach was confirmed in ovarian tumour xenograft studies, revealing a promising avenue for future cancer therapies.

[1] Ruhr University Bochum, Faculty of Medicine, Department of Haematology and Oncology, Institute of Molecular Oncology and Experimental Therapeutics, Marien Hospital Herne, Herne, Germany. [2] Central Animal Laboratory, University Hospital Essen, University of Duisburg-Essen, Essen, Germany. [3] Institute of Cell Biology, Cancer Research, University Hospital Essen, University of Duisburg-Essen, Essen, Germany. [4] Visceral Surgery Department, Marien Hospital Herne, Ruhr University Bochum Medical School, Herne, Germany. [5] Institute of Pathology, Ruhr University Bochum, Bochum, Germany. [6] Department of Life and Environmental Sciences, University of Cagliari, Cittadella Universitaria di Monserrato, Monserrato, Italy. [7] Department of Anesthesia, Intensive Care, Pain and Palliative Medicine, Marien Hospital Herne, Ruhr-University Bochum Medical School, Herne, Germany. ✉email: david.diaz-carballo@marienhospital-herne.de

Toll-like receptors (TLRs) play a pivotal role in innate immunity by recognising pathogen-associated molecular patterns[1]. TLR3, 7, 8 and 9 are located in the endosome and collectively play a critical function in the detection of viral signatures. TLR7/8 are phylogenetically and structurally very similar and both can recognise GU-rich sequences in single-stranded RNAs (ssRNAs) from viruses[2]. TLR7/8 are coupled to the adaptor protein MyD88 which activates downstream NF-κB driven-genes[3]. NF-κB plays an important role in the initial formation and progression of malignancies[4]. It triggers the expression of apoptosis inhibitors, growth and angiogenesis factors, cyclins, as well as metalloproteinases involved in tumour invasiveness[5]. TLR7/8 are also involved in a plethora of intracellular signalling cascades which culminate in the gene expression of pro-inflammatory cytokines and chemokines[6]. Both receptors are also widely expressed in several types of cancer and therefore, agonists of these TLRs are currently of interest as antitumoural compounds[7–11].

Among the proteins aberrantly expressed in malignant cells, those derived from human endogenous retroviruses (HERVs) represent a vast group with a strong link to cancer cell biology, which makes them potential targets for therapy[12,13]. HERVs constitute a broad class of retroviral genetic elements, which were integrated into the host genome and are vertically transmitted in a Mendelian form to the progeny. While intact HERVs possess a typical 5′-LTR-gag-pol-env-LTR-3′ retroviral structure, HERVs in humans (and other apes) have lost the capability of horizontal transmission due to epigenetic repression and/or the accumulation of mutations, which eliminated their infective capacity. Yet, some HERV genes have retained intact ORFs that via complementation may create transducing particles which form bioactive quasi-species[12,14–18]. Some of these endoretroviral elements have also been linked to increased tumour cell heterogeneity[19] and suggested to promote carcinogenesis through the very same function they perform in placenta physiology—fusogenicity and immunosuppression—to stimulate uncontrolled cell fusion and abrogate the anti-oncogenic cytolytic immune response[20,21].

An Env-coding HERV sequence harboured by the human genome and expressed in the placenta (locus 19q13.41) is HERV-V. It is more ancient than the two human syncytins as it was acquired by the primate lineage more than 45 million years ago, and is found in both New and Old World Monkeys[22,23]. This provirus has undergone post-integrative duplication which resulted in the emergence of HERV-V1 (producing the C-terminally truncated protein EnvV1) and HERV-V2 (coding for the full-length EnvV2), which are ~34 kb apart and show high nucleotide identity. While behaving like a syncytin in Old World Monkeys, EnvV2 has lost its fusogenicity in hominoids through a slow, still-ongoing selective process which, however, did not affect its immunosuppressive activity[23].

The sole expression of HERVs is not an adequate stimulus for recognition by TLR7/8, not even overexpression observed in cancer cells refractory to cytostatic therapy. This indicates that a minimum level of ssRNA with retroviral signatures does not exist, which would be needed by TLR7/8 to render cells susceptible to trigger apoptosis. Even when the HERV RNA pool is overexpressed through subtoxic isodoses of HDAC inhibitors (HDACis), no cell death is achieved. Therefore, the cellular concentration of HERV RNA is not sufficient to mimic an infection, and more factors may interfere in this context. In nature, this phenomenon is observed in viral infections in which cells do not adequately activate the innate immunity, i.e. TLR7/8 cells are not stimulated.

To overcome this problem, we extrapolated the concept of "shock and kill" (SaK) therapy to cancer treatments. SaK is now the dominant strategy aimed at eliminating HIV from tissue reservoirs and basically describes a two-step intervention: first, latency-reversing agents are used to reactivate latent HIV (the "shock") and secondly, the "kill" phase of those virus-expressing cells using neutralizing monoclonal antibodies or TCAR against viral structures[24–26].

In this context, HDACis as for example vorinostat and romidepsin were employed as "shock" agents. In ovarian cancer (OC) models, we observed a drastic, dose-dependent expression of HERV-V1/V2 envelope genes under the influence of HDACis. Their role in cancer cell biology is fully unknown, but they have been detected or overexpressed in malignant tumours and not in regular adjacent tissues. For the "kill" phase, instead of components of the humoral or cellular immunity, we involved the innate immunity by employing TLR7/8 agonists (TLR7/8as). Besides their antiviral properties, they also show a robust antitumoural activity which is not fully understood[7,8,10,11]. This combination acts synergistically and effectively induces cell death at subcytotoxic doses. Steady-state analyses of specific canonical signal transduction cascades like TLR7/8-NF-κB, Akt/PKB, Ras-MEK-ERK and apoptotic pathways further revealed consistency with bidirectional interaction of the two substance classes. The effects observed in vitro were mirrored in ovarian carcinoma xenograft models, giving further credence to this strategy as a potential anticancer therapy.

In summary, our data indicate that the combination of HDACis acting as Latency Reversing Agents (LRAs) together with TLR7as may represent a novel, unprecedented therapeutic strategy to eliminate cancer cells. The fact that some drugs used in this study are FDA-approved might open up a new avenue for clinical trials to define therapeutic efficacy and possible adverse effects.

## Results

**HERV expression in OC cell lines, ovarian tumours and surrounding tissues.** A range of HERV elements was detected both at the RNA and protein level in a panel of OC cell lines and primary tumour cells (Fig. 1, Supplementary Tables 1 and 2). Interestingly, some HERV envelope proteins were expressed beyond basal levels in SKOV3[CP] carboplatin-resistant OC cell subline and in OvCa236 ascites-derived primary OC cells obtained from heavily treated patients (Fig. 1a). Expression of HERV-V1/V2 protein in particular was mainly confined to the tumour areas (Fig. 1b) and considerably less detectable in healthy tissue and the cancer-surrounding stroma (Fig. 1b).

The fact that these retroviral elements were overexpressed in chemorefractory cancer cells provided a first hint that they may also be susceptible to chemical activation by HDACis, which are known to act as LRAs for genome-integrated retroviruses like HIV[27].

**HDACis are potent de-repressors of HERV-V1/V2 transcription.** In line with the above hypothesis, we investigated the role of HDACis in driving HERV transcription and specifically the transcription of genes, which encode for envelope proteins. For this purpose, we tentatively selected romidepsin and vorinostat, two therapeutics that are currently in clinical use and known to be potent HIV anti-latency drugs in the context of the SaK approach. Their $IC_{50}$ values were determined over 72 h in different OC cell lines and primary cells isolated from OC ascites (Fig. 2a, Supplementary Fig. 1). Romidepsin and vorinostat were found to be highly cytotoxic at concentrations of ~4 nM and 1.5 μM, respectively. These values were used as reference in all subsequent biological experiments.

Using fractions of the $IC_{50}$ values, we analysed the transcriptional activation of a panel of HERV elements (Supplementary

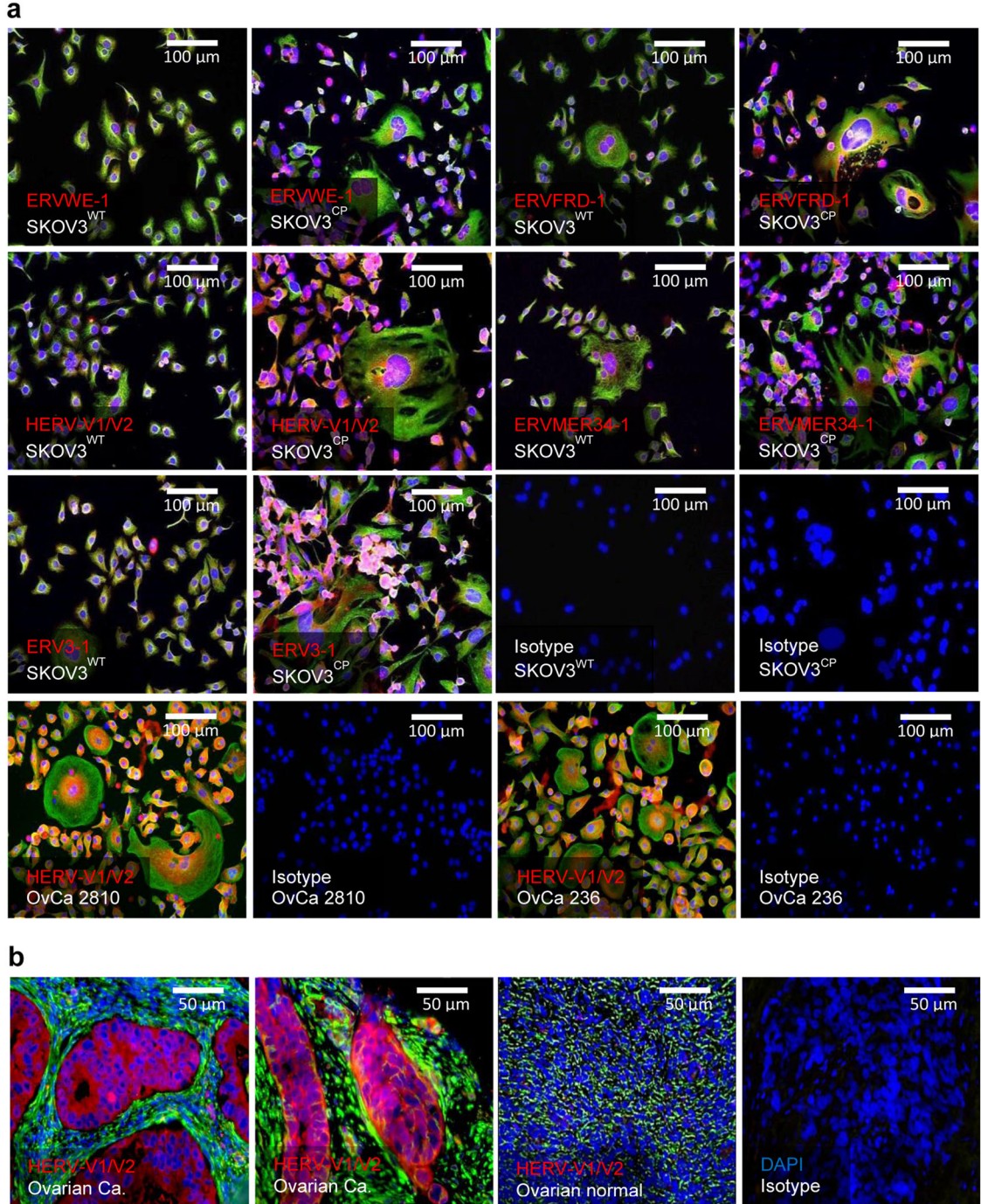

**Fig. 1 Expression of HERV proteins in ovarian carcinoma. a** ERVWE1 (syncytin 1), ERVFRD-1 (syncytin 2), HERV-V1/V2, ERVMER34-1 and ERV3-1 are expressed in the SKOV3$^{WT}$ ovarian carcinoma cell line. Of note, in the carboplatin-refractory SKOV3$^{CP}$ subline, these envelope proteins are remarkably overexpressed, establishing a correlation between HERV expression and high chemotherapy resistance. Isolated ascites cells deriving from heavily carboplatin-treated ovarian carcinoma are highly positive for HERV-V1/V2 envelope proteins. **b** In tissue paraffin sections from ovarian carcinoma of epithelial origin, HERV-V1/V2 expression is mainly confined to the tumour area. It is low in the tumour stromal tissues and almost absent in regular ovarian tissues (IHC analysis, magnification ×20). Results are representative of $n = 10$ analysed specimens.

Table 1) after 24 h of drug exposure. It was found that some HERV genes were effectively de-repressed (activated) by our HDACis among which *HERV-V1/V2* showed greater post-exposure transcriptional activity than the other HERVs analysed. This effect was consistent in almost all cancer cells, which we looked at (Supplementary Fig. 2).

Seeing that some HERVs could be transcriptionally activated by HDACi treatment, we asked the question whether this effect

might be exploited in analogy to SaK to induce selective cancer cell death in two ways: by forcing the expression of HERV envelope proteins in the cell membrane and targeting them with specific antibodies or CART systems, and second, by recruiting the cell's own innate immune response via agonization of TLR7/8 receptors, which recognise viral signatures. We therefore determined the IC$_{50}$ of the TLR7/8as imiquimod and vesatolimod which are currently in clinical use (Fig. 2b, Supplementary

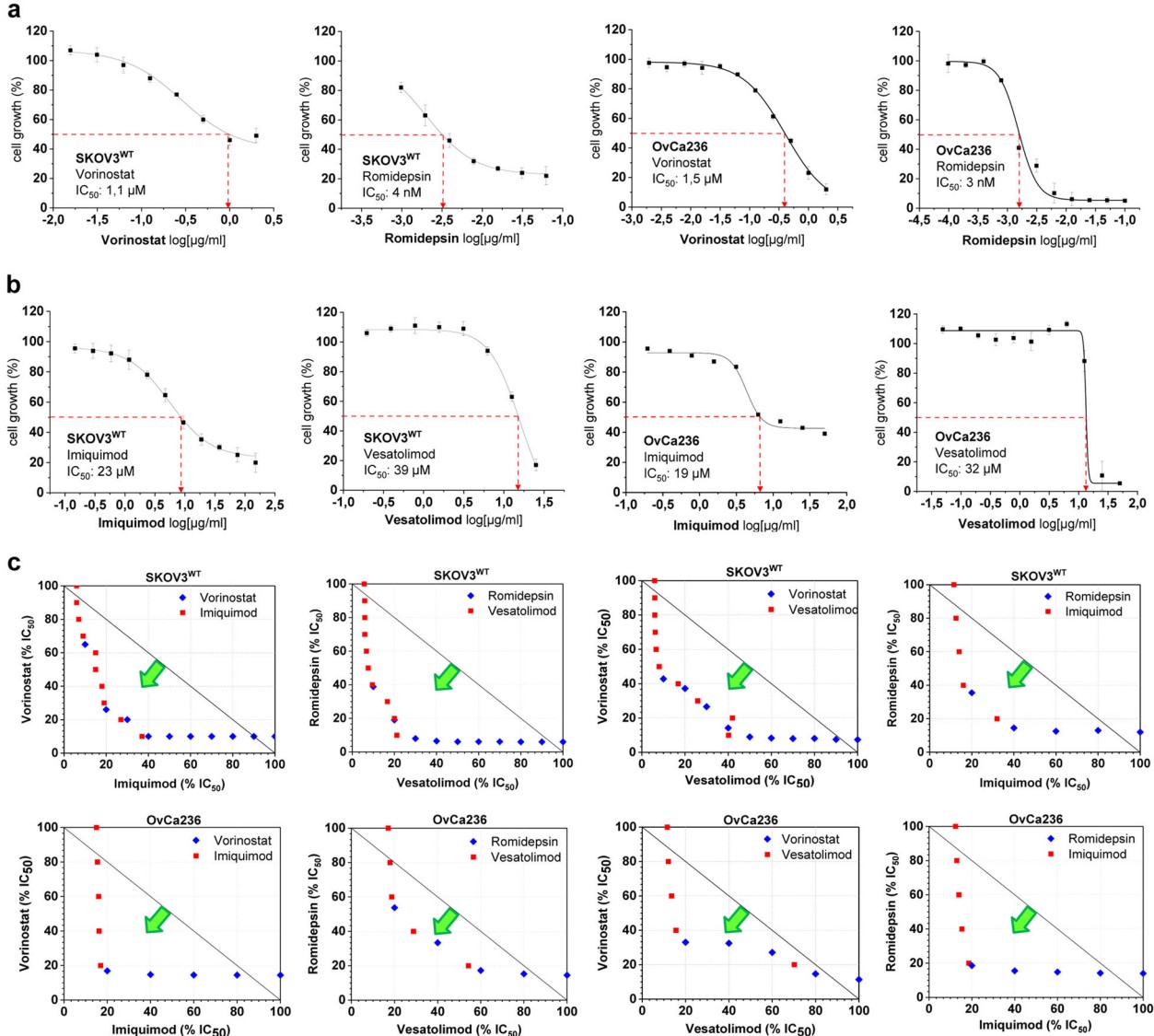

**Fig. 2 Cytotoxicity and interaction of HDACis and TLR7as in ovarian carcinoma cells. a** The cytotoxicity of the HDACis vorinostat and romidepsin in SKOV3[WT] and OvCa236 primary ovarian carcinoma cells was determined using the MTT proliferation assay after 72 h of drug exposure. Vorinostat and romidepsin are very cytotoxic in ovarian carcinoma cells ($IC_{50}$ ~ 4 nM). **b** The TLR7as imiquimod and vesatolimod were tested in the same manner as the HDACis proved less cytotoxic in these carcinoma cell types, with $IC_{50}$ values between 23 and 32 μM. These values were employed as reference in all subsequent biochemical experiments. The graphs are representative of $n = 5$ experiments. **c** HDACi/TLR7a interaction was analysed using isobolographic methods in SKOV3[WT] and OvCa236 primary ovarian carcinoma cells isolated from ascites. The combination of HDACis and TLR7as produces a synergistic effect regardless of pairing (green arrows). Results are representative of $n = 5$ experiments.

Fig. 3a). Compared to HDACis, both agents proved less cytotoxic in cancer cells, with an $IC_{50}$ of around 40 μM in each case.

**Synergistic cytotoxic effects of HDACis and TLR7/8as in OC cell lines and primary tumour-derived cells**. To obtain a first hint of the bidirectional cytotoxic activity of HDACis and TLR7/8as in OC cell lines and primary tumour cells, we performed an isobolographic analysis which combined the two drug classes (Fig. 2c, Supplementary Fig. 3b). Our isobolograms reveal that the 'HDACi and TLR7/8a combination treatment' (HTCT) always produced a synergistic cytotoxic effect, regardless of how the two substance classes were combined. More importantly, the HTCT resulted in a synergistic cytotoxic effect in SKOV3[CP] subline, which is highly resistant to carboplatin (Supplementary Fig. 3c).

**HTCT is a potent, selective inducer of intrinsic apoptosis**. Our studies on apoptotic pathways in OC cells revealed that the synergistic cytotoxic interaction between HDACis and TLR7/8as specifically triggers intrinsic apoptosis, confirmed in particular by detecting the cleavage of caspase 9. In addition, anti-apoptotic executors like Bcl-xL were downregulated at the protein level, thus favouring apoptotic processes[28] (Fig. 3a, Supplementary Figs. 4 and 5).

Moreover, immunochemical analysis with ICC, FACS and WB performed in SKOV3[WT] and primary OC cells showed that HTCT led to extensive PARP (poly-ADP ribose polymerase) cleavage (Fig. 3a–c, Supplementary Figs. 4 and 5). Dose-dependent expression of cleaved caspases 3 and 9 as well as PARP was observed in vorinostat and imiquimod treated SKOV3[WT] cells (Supplementary Fig. 6a). The level of apoptosis was identical in the cell population, which remained attached to

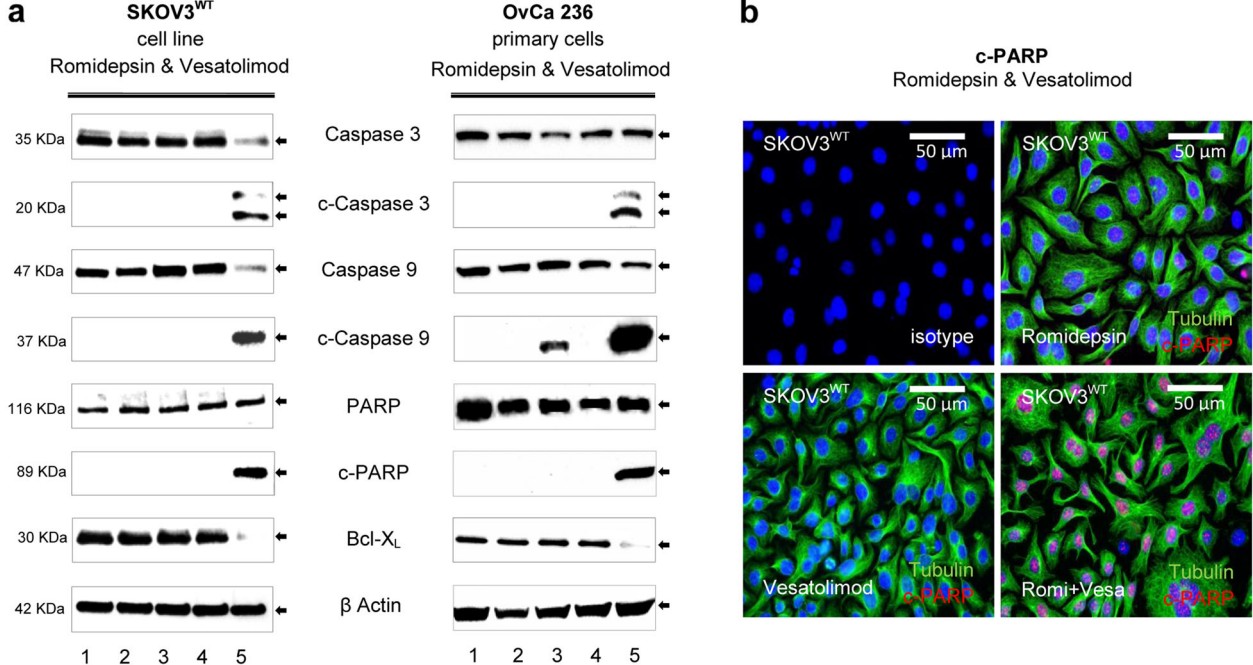

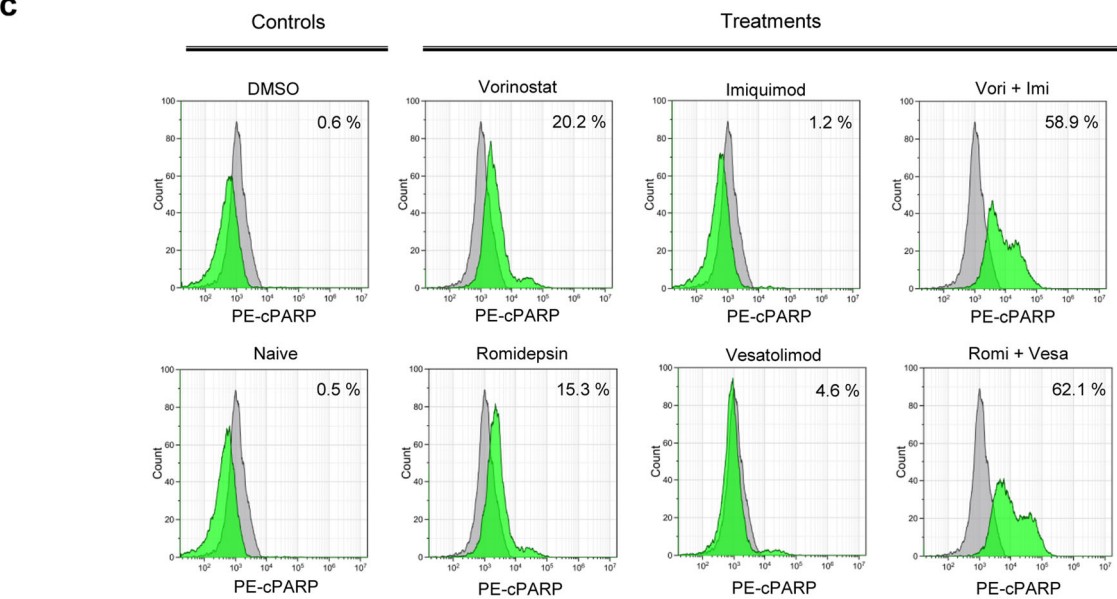

**Fig. 3 HTCT induces selective intrinsic apoptosis. a** After incubation of SKOV3[WT] and OvCa236 cells with 1× IC$_{50}$ of HDACi, TLR7a and their combinations for 24 h, apoptosis mediators were analysed by western blotting, revealing the cleavage of central apoptosis pathway mediators such as caspase 3 and caspase 9, the latter being a determinant for the intrinsic pathway. In congruence, the specific cleavage of PARP after HTCT underlines the specificity of the interaction of the two substance groups. The anti-apoptotic protein Bcl-xL is downregulated by HTCT but not by the individual drugs. **b** c-PARP is predominantly detected (ICC) in SKOV3[WT] cells incubated with romidepsin and vesatolimod combined but not with vesatolimod alone (0.5× IC$_{50}$). Magnification ×40. **c** FACS analysis for cleaved PARP shows that apoptosis is triggered by HTCT but not by the individual drugs. Isotype: grey and sample: green. Results are representative of $n \geq 3$ experiments.

the culture surfaces and in the detached, floating populations (Supplementary Fig. 6b). Interestingly, HTCT showed no synergistic effect in triggering apoptosis of PBMCs derived from healthy individuals (Supplementary Fig. 6c).

**HTCT induce an impairment on HERV-V1/V2 protein synthesis**. HTCT rendered in strong synergistic effect, which was correlated with the decrease of HERV-V1/V2 proteins in OC cells, as judged by both FACS and Western blot analysis (Supplementary Fig. 7a, b). Thus, the reduced HERV-V1/V2 protein levels occasioned by the drug combinations might be coupled from cell death if we assume that the remaining attached populations are on their way to apoptosis as judged by ICC analysis (Fig. 3b). The effect of HDACis on protein synthesis is uncoupled

from the upregulation of this gene at transcriptional level (Supplementary Fig. 8).

**TLR7 and HERV-V1/V2 gene ablation with CRISPR/Cas9 significantly reduces apoptosis.** In order to determine whether the observed pro-apoptotic effects were mediated by retroviral RNA signatures via TLR7, we separately ablated both *TLR7* and *HERV-V1/V2* in SKOV3$^{WT}$ cells using CRISPR/Cas9, without selecting for the ablated populations (Supplementary Figs. 9 and 10). Gene ablation was performed by targeting mRNA for TLR7 (MN_016562.4, Supplementary Fig. 9a) or HERV-V1 (NM_152473.2, Supplementary Fig. 10a) using specific gRNA designed with an IDT tool (https://eu.idtdna.com/site/order/designtool/index/CRISPR_CUSTOM) and checked against the whole human genome to determine on/off-targets with the Basic Local Alignment Search Tool (BLAST). The ablation effectiveness was confirmed by T7 digestion and qPCR (Supplementary Figs. 9b, c and 10b, c). In addition, the comparative protein expression for HERV-V1/V2 in wild-type and KO cells was assessed (Supplementary Fig. 10d). In the SKOV3$^{HERV-V1/V2KO}$ cells, the amplicons for HERV-V1/V2 were ~50% less expressed after HDACis exposure compared to the wild-type cells (Supplementary Fig. 11a).

As shown above, HTCT induced a strong apoptotic response in several cell systems. In contrast, apoptosis in SKOV3$^{TLR7KO}$ cells was noticeably less pronounced as determined by PARP cleavage (Fig. 4a, b), indicating that TLR7 is a pivotal mediator of post-treatment cell death. Similar to TLR7, we detected a global 50% reduction in apoptotic cells in the HERV-V1/V2 knockout cultures, suggesting that the products of these genes are indeed mediators of apoptosis (Fig. 4c–e). A microscopic investigation confirmed that the SKOV3$^{HERV-V1/V2KO}$ subline was insensitive to romidepsin/vesatolimod combination treatment (Supplementary Fig. 11b).

The significant reduction in apoptotic cells after *TLR7* and *HERV-V1/V2* gene ablation validates the involvement of these elements in mediating cell death driven by HTCT, and suggests a possible extrapolation of the SaK concept to cancer therapy.

**Do other HERV elements play into the SaK effect?** From dose-dependent studies, we observed that *HERV-V1* looks more HDACis-activatable than the truncated *HERV-V2* form (Fig. 5a). Although we had established that the *HERV-V1/V2* gene products act as TLR7 substrates, a subsequent qPCR analysis hinted that HDACi treatment de-represses further HERV elements in OC cells (Supplementary Fig. 2). In order to study this issue in detail and pinpoint the possible contribution of other HERVs, we performed a ChIP-seq in SKOV3$^{WT}$ cells exposed to romidepsin and vorinostat. Analysing the histone 3 (H3) acetylation patterns after the treatment, we concluded that H3AcK9, which is known to be involved in transcriptional activity, was the most suitable candidate for chromatin immunoprecipitation (Supplementary Figs. 12 and 13).

To check the acetylation levels at the positions harbouring HERV loci and assess their variation after the treatment with HDACis, the univocal genomic coordinates of about 3280 HERV loci have been compared with the position of the H3AcK9 peaks identified by ChIP-seq analysis (Supplementary Fig. 14, Supplementary Data 1). Overall, in untreated cells, around 0.7% of peaks corresponded to 1233 individual HERV loci (37.6% of the whole dataset), and the number of co-localised peaks was increased of 2.3–2.5-fold after the treatment with HDACis, mapping to a percentage of HERV loci from 48 to 60% of the total dataset (Supplementary Fig. 14). This first analysis confirmed that HDACis are able to modulate the acetylation of repetitive

elements, including HERV sequences. Subsequently, we considered only the differential peaks (diffpeaks), i.e. the peaks showing significant variation after the treatment with vorinostat and/or romidepsin. Interestingly, even if more than half of HERV loci co-localised with acetylation peaks after HDACi treatment, only a minority of them (0.9–1.9%) co-localised with diffpeaks, showing mostly a reduction in their acetylation levels (Supplementary Fig. 14, Supplementary Data 1). Moreover, the exposure to different HDACis led to the variation of different subset of HERV-loci acetylation, even if a proportion of HERV sequences (21/82) were common to both treatments (Supplementary Fig. 14). Diffpeaks co-localised HERV loci could be classified into 24 HERV groups and included the *ERV-V2* locus (Supplementary Fig. 14).

We focused then on the 11 HERV loci which co-localised with diffpeaks showing an increase in acetylation levels, hence indicating an increased transcriptional activation after HDACi treatment (Table 1). Remarkably, diffpeaks corresponding to the *ERV-V2* locus were identified in all samples and were the most significantly upregulated in both vorinostat and romidepsin treatment (Table 1, Supplementary Fig. 14). This was in part confirmed by ChIP-qPCR in different OC cells (Supplementary Fig. 15, Supplementary Table 3). Beside *ERV-V2*, only the *HML8* sequence at locus 2q11.2 showed increased acetylation levels in all treated samples, independent from the HDACis used, but having lower statistical significance (Table 1, Supplementary Data 1). Given that *ERV-V2* locus showed a specific increase of acetylation levels following HDACi stimulation, we assessed the position of diffpeaks within its retroviral structure and compared it with respect to the corresponding peak in untreated control (Fig. 5b). The analysis revealed that the treatment with HDACis did not only account for the increase of *ERV-V2* acetylation, but also led to the shift and concentration of acetylation peaks at the 5′ region of the HERV provirus. This observation allows us to hypothesize that HDACi treatment could not only increase *ERV-V2* transcriptional activation, but also modify its pattern of expression, possibly favouring HERV-specific promoter activity from *ERV-V2* LTR or an upstream solitary LTR43 element, and eventually leading to alternative splicing (Fig. 5b). This scenario is also supported by the presence of promoter and proximal enhancer-like signatures that are collinear with HDACi-associated diffpeaks.

**Studies on TRL7 signalling validate the mechanism of SaK in cancer cells.** TLR7 transduces signals by recruiting the MyD88 adaptor protein, which in turn communicates with the transcription factor (TF) NFκB via a sequence of transducers. NFκB helps initiate the transcription of a plethora of genes implied in the inflammation network, cell survival and differentiation, and it is also found upregulated in cancers[29]. We observed that HTCT led to the functional disruption of this pathway (Fig. 6a). We incubated different carcinoma cell lines with both drug classes simultaneously (0.5× IC$_{50}$ for 24 h) and studied how the RNA and protein levels of pivotal transducers were affected. The transcription of TLR7, as the first player in this pathway, was not negatively affected by HTCT (Fig. 6b). In contrast, protein synthesis of MyD88 was selectively impaired, suggesting a disruption of the downstream signalling switch of this transducer (Fig. 6a, c, Supplementary Table 4).

Given that SARM1 has been proposed to be an adaptor protein for TLR7[30], we investigated its role in TLR7 downstream signalling and in mediating apoptosis in response to HTCT. To address this question, we ablated SARM1 using CRISPR/Cas9 in SKOV3$^{WT}$ cells, incubated the cells simultaneously with romidepsin and vesatolimod (1× IC$_{50}$) and checked for apoptosis

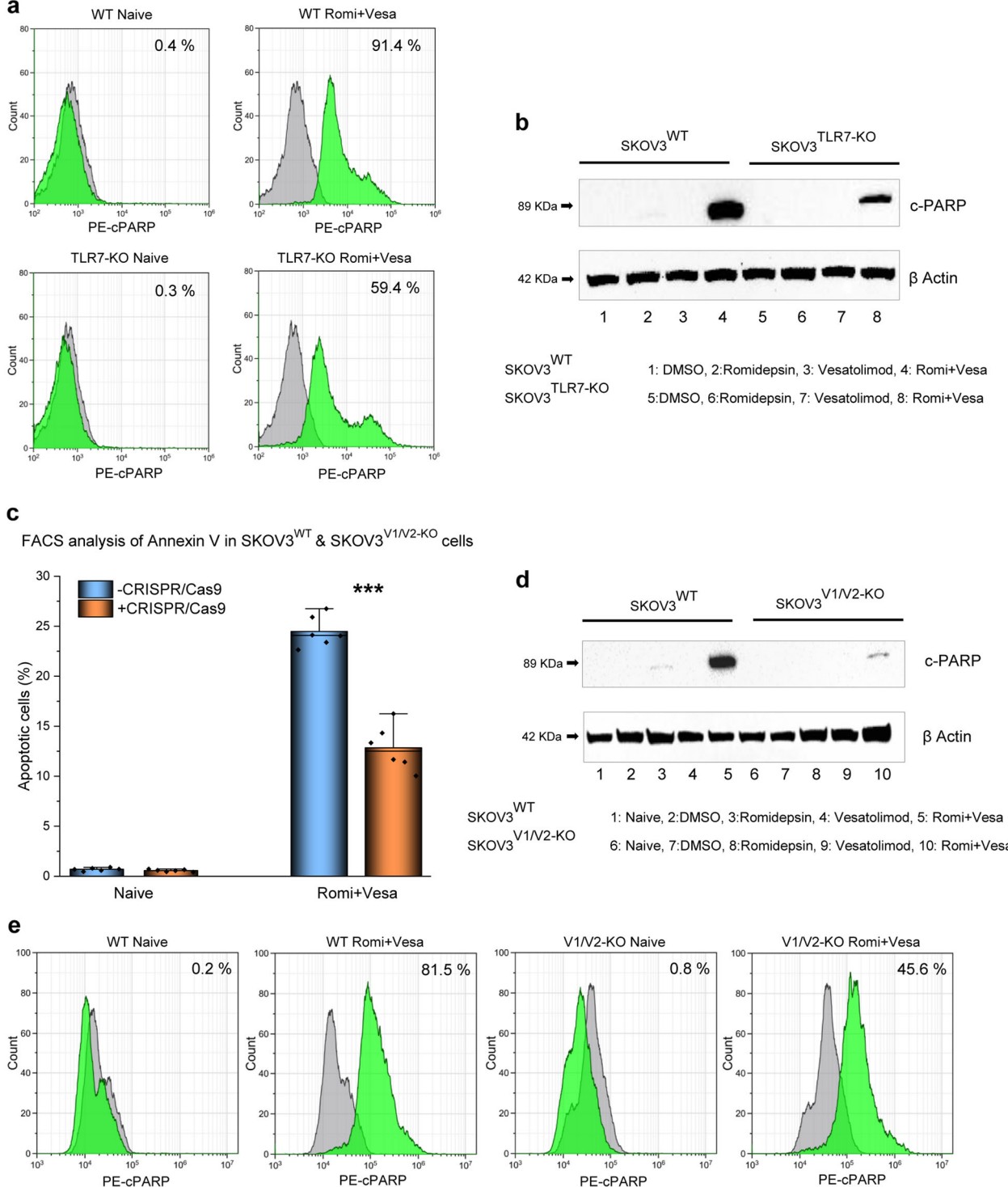

**Fig. 4 TLR7 and HERV-V1/V2 are direct mediators of HTCT-induced apoptosis. a** CRISPR/Cas9 ablation of *TLR7* impairs HTCT-induced apoptosis in SKOV3 cells, indicating that apoptosis is mediated by TLR7 in this system (determined by PARP cleavage FACS). Isotype: grey and sample: green. **b** Western blot analysis for c-PARP, demonstrating the same effect. **c** *HERV-V1/V2* ablation reveals the role of the *HERV-V1/V2* gene product as substrate for TLR7 (cytometric FACS measurement of annexin V); *$p < 0.05$, **$p < 0.01$, ***$p < 0.001$ (paired Student's *t* test; two-tailed). **d** HERV-V1/V2-ablated SKOV3 cells are less susceptible to HTCT-induced apoptosis than the non-ablated parental cells (western blot). **e** About 36% reduction in apoptosis is observed in SKOV3[HERV-V1/V2KO] cells (FACS analysis of PARP cleavage). Isotype: grey and sample: green. Results are representative of $n \geq 3$ experiments.

using c-PARP as a sentinel for cell death. Although SARM1 protein tended to be reduced after HTCT, the difference in apoptosis between ablated and normal cells was not significant (Supplementary Fig. 16), indicating that SARM1 is not likely to play a role as a TLR7 adaptor in this cell line.

Following romidepsin/vesatolimod combination treatment, the protein synthesis of undigested NFκB1 (p105 form) was reduced, suggesting that I-κBα was hypophosphorylated, the NFκB/I-κBα complex did not dissociate, and NFκB1 was not transcribed; this effect was confirmed a posteriori with immunotechniques

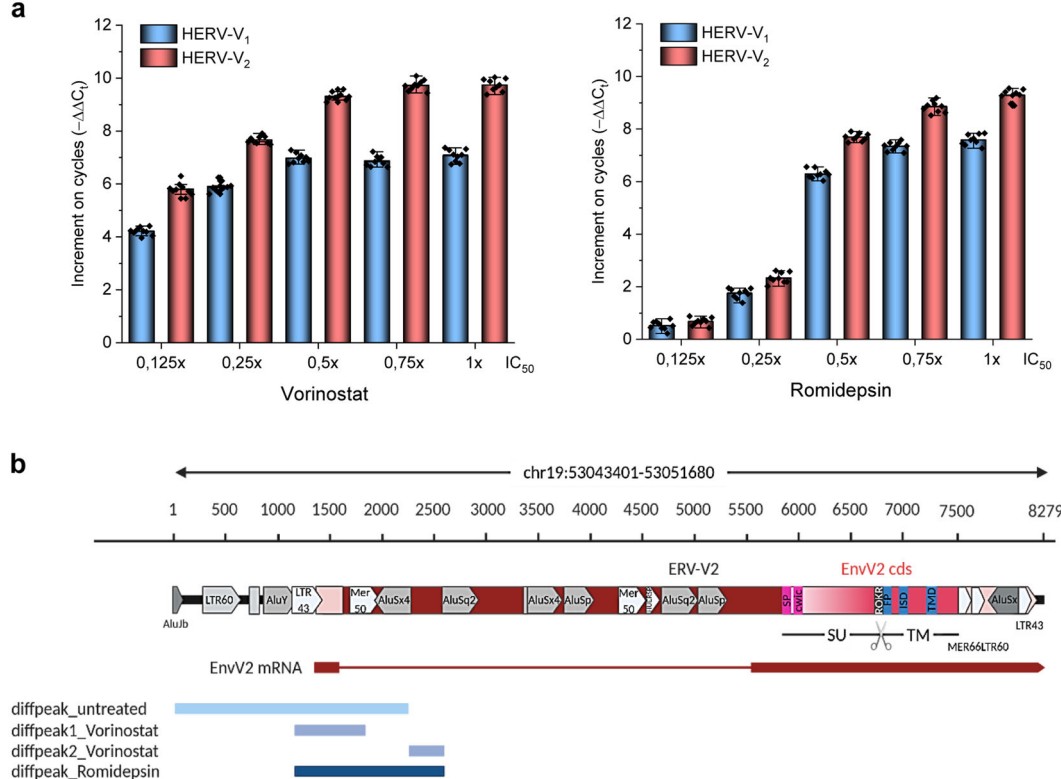

**Fig. 5 ChIP-seq analysis after HDACi treatment in SKOV3^WT. a** Dose-dependent activation of the *HERV-V1/V2* genes in SKOV3^WT cells exposed to different IC$_{50}$ fractions of HDACi for 24 h. **b** The *HERV-V2* locus (chr19:53044740–53051680) is highlighted by a maroon arrow, along with the mRNA encoding for the EnvV2 protein. Putative LTR sequences are highlighted in peach colour. The EnvV2 coding sequence (cds) is also shown, with the annotations for surface (SU) and transmembrane (TM) subunits: hydrophobic signal peptide (SP), CWIC motif involved in SU and TM interaction (consensus: CXXC), RKQR furin cleavage site separating SU and TM portions (consensus: RXKR), fusion peptide (FP), putative immunosuppressive domain (ISD), and transmembrane domain (TMD). LTR and non-LTR retrotransposon secondary integrations are highlighted by white and grey arrows, respectively. The positions of co-localised diffpeaks before and after HDACi treatment are shown. Promoter (1145-1434) and proximal enhancer (1588-1751 and 2265-2490) like sequences referring to the whole chromosome region shown as predicted by ENCODE registry of candidate cis-regulatory elements.

**Table 1 HERV loci colocalise with diffpeaks, indicating increased acetylation after HDACi treatment.**

| HERV loci info | | | | | | diffpeak Log$_{10}$ (*p*-value) | |
|---|---|---|---|---|---|---|---|
| **Chr** | **Start** | **End** | **Strand** | **ID** | **Group** | **Vorinostat** | **Romidepsin** |
| 19 | 53044740 | 53051680 | + | ERV-V2 | ERVV | 5.32 | 8.78 |
| 19 | 5548587 | 5553253 | + | 4594 | HERVH | — | 5.98 |
| 1 | 117815373 | 117821863 | − | 6028 | HERVS | 3.94 | 5.51 |
| 14 | 92622417 | 92629608 | − | 4215 | HERVIP | 4.41 | — |
| 11 | 121632566 | 121643491 | − | 3656 | HERVH | 4.27 | 3.17 |
| 5 | 10716252 | 10725049 | + | 1756 | HERVL | 3.85 | — |
| 2 | 100361988 | 100365940 | − | 2q11.2 | HML8 | 3.52 | 3.70 |
| 4 | 20194383 | 20199906 | + | 1335 | HERVH | — | 3.59 |
| 3 | 94647178 | 94651193 | + | 3q11.2 | HML7 | — | 3.50 |
| 19 | 21786150 | 21792221 | − | 19p12b | HML6 | 3.09 | 3.34 |
| 1 | 185578774 | 185596328 | − | 6125 | HML5 | 3.10 | — |

Differential peak (diffpeak) *p*-values are depicted as the mean of two independent CHIP-seq experiments.

(Fig. 6a). It means that, after treatment, the full-length form of the canonical transcription factor NFκB1 cannot translocate into the nucleus and perform its role as TF, which was corroborated with a luciferase system containing a consensus sequence for this TF coupled to a firefly luciferase as reporter (Fig. 6d). Our experiment established that NFκB is not functional even if stimulated with TNFα and hence cannot initiate the transcription

of genes which are in part governed by it, including the gene for NFκB itself. A critical NFκB target gene is Bcl-xL[31], which was selectively reduced at the protein level after HTCT (Fig. 3), confirming the disruption of this transcription factor. IL8 appeared to be less affected, as its transcription was down-regulated under the influence of either HTCT in OvaCar3 and OvCa2810, but less so in SKOV3^WT. What we observed in

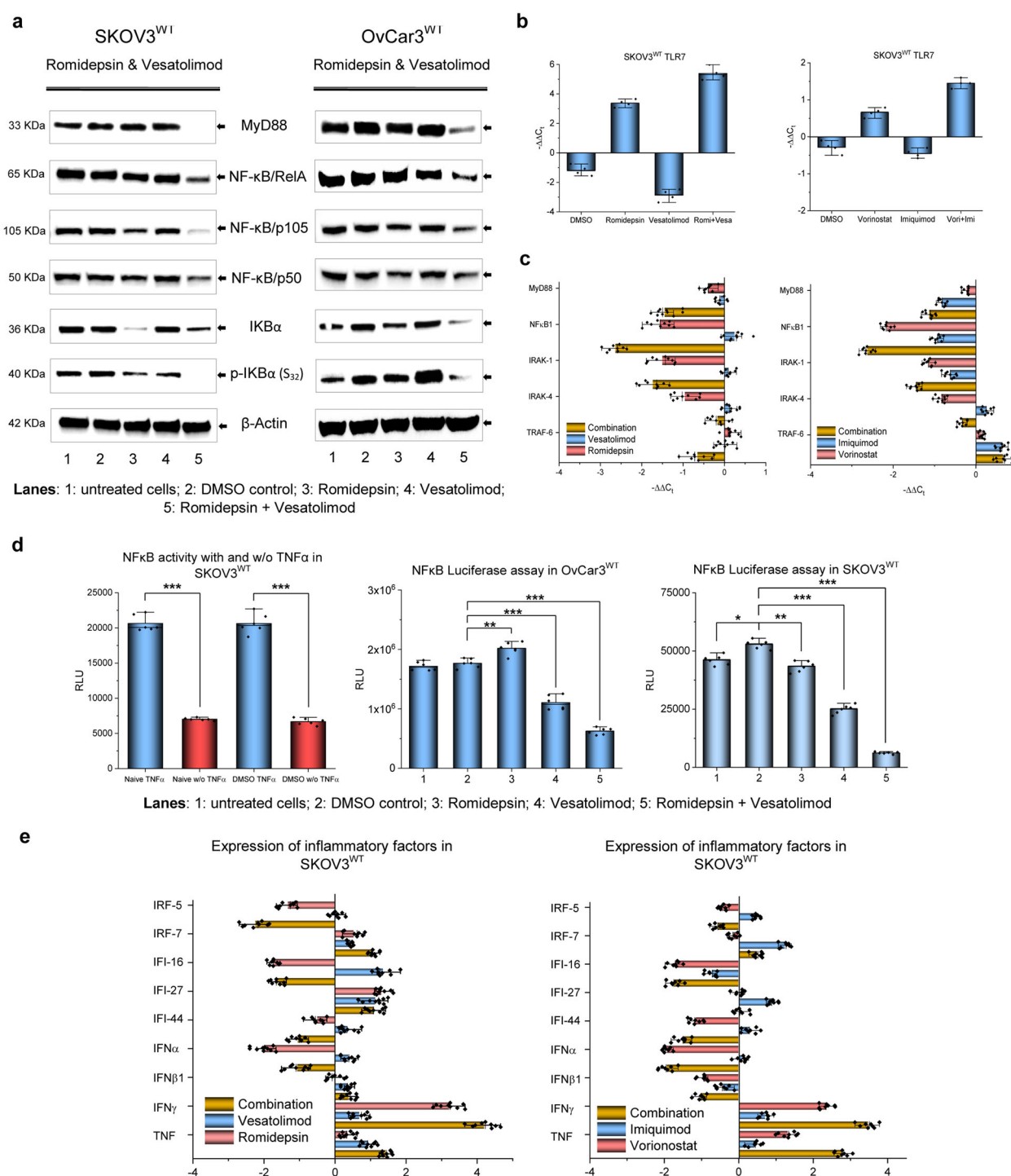

**Fig. 6 Influence of HDACi and TLR7a on TRL7 signalling in OC cells. a** HTCT affects signalling downstream of TLR7 up to the last effector, i.e. NFκB, in the OC cell lines SKOV3$^{WT}$ and OvCar3$^{WT}$. MyD88 and NFκB protein synthesis is selectively repressed. Note the hypophosphorylation of the kinase IKBα, which is responsible for the degradation of the transcription factor NFκB. **b** Influence of HDACis, TLR7/8as and their combination on TLR7 transcripts in SKOV3$^{WT}$ cells (qPCR). The combination treatment upregulates TLR7 transcripts. **c** Effect of HTCT on TLR7 signalling at the mRNA level in SKOV3$^{WT}$ cells. **d** Reduced hypophosphorylation of p105 after drug incubation inactivates NFκB, determined using the NanoLuc® assay containing a consensus for NFκB and stimulating both cell lines with TNFα to amplify the system signals. *$p < 0.05$, **$p < 0.01$, ***$p < 0.001$ (paired Student's $t$ test; two-tailed). **e** Impact on inflammatory signalling: transcript levels of a panel of inflammatory mediators in SKOV3$^{WT}$ differed when treated with single drugs, but were very similar to HTCT. In contrast, both IFNγ and TNFα were significantly upregulated. Results are representative of $n \geq 3$ experiments.

OvarCa3 and OvCa2810 is a compensatory homoeostatic behaviour of the axis: IL downregulation vs. CXCR1 upregulation (Supplementary Fig. 17). IL8 is a rather promiscuous interleukin, owing to its ability to dock several chemokine receptors, the most prominent being CXCR1 followed by CXCR2. IL8 is upregulated in several cancer types, including OC, and is subject to transcriptional regulation by NFκB and other TFs[32].

We also examined the impact of HTCT on inflammatory signalling at the RNA level in SKOV3[WT], which are also dependent on NFκB. As depicted in Fig. 6e, the expression profile of some inflammatory mediators was very similar for the two combinations but differed when the drugs were applied individually (Supplementary Table 4). IFNγ and TNFα, in contrast, were substantially upregulated in all cases. These inflammatory factors were also analysed partially ex vivo using PBMCs from healthy individuals (Supplementary Fig. 18).

**HTCT disrupts other signal transduction pathways**. In an effort to further clarify the mechanism(s) leading up to the apoptotic response upon HTCT, we investigated canonical signal transduction pathways such as Akt/PKB and Ras-MEK-ERK in different OC cell lines. These pathways are tightly interconnected with NFκB signalling, as I-κBα, for example, is known to be phosphorylated via Akt/PKB[33,34].

Western blot analysis of specific transducers revealed that all drug combinations reduced the survival protein Akt at the protein level and suppressed its phosphorylated form entirely, while imiquimod alone induced its hyperphosphorylation (Fig. 7a, Supplementary Figs. 19 and 20). The dephosphorylated and phosphorylated forms of β-catenin and c-Myc were downregulated at the protein level, similar to Akt. Taken together, Akt activation was fully disrupted by all analysed HTCT.

As previous works have reported that some HERVs are associated with the activation of Ras-MEK-ERK[35] cascade, we investigated the effect of HTCT on this pathway. In fact, a disruption was seen in this cascade. While ERK1/2 and MEK1/2 protein levels were not reduced after treatment with either HDACis, their phosphorylated forms were depleted, while the opposite effect was observed after treatment with the TLR7 agonists. In cancer cells, MyD88 constitutively inhibits MKP3 phosphatase activity on MEK1/2, which results in constitutive MEK1/2 activation. We showed that MKP3 protein expression was not affected by any of the analysed drug interventions. Since MyD88 was downregulated by each of the HTCTs, we expected MEK1/2 to be dephosphorylated. Yet, the dephosphorylation of its downstream transducer under HDACi treatment suggested that other mechanisms might be at play that go beyond the interaction between MyD88 and MKPs. Figure 7b depicts global disruption points of the TLR7-NFκB, Akt/PKB, and Ras-MEK-ERK cascades upon HTCT in OC cells.

**Synergistic anticancer activity in an SKOV3[WT] xenograft mouse model**. The cell lines SKOV3[WT] and OvCar3[WT] generate well-differentiated adenocarcinomas in xenograft nu/nu mice. We studied the impact of romidepsin and vesatolimod administered either individually or in combination on tumour growth and metastasis formation in an SKOV3[WT] xenograft mouse model (Fig. 8). A marked antitumoral effect was observed after four rounds of treatment. While romidepsin and vesatolimod produced an antitumoral effect when administered alone, their combination mirrored the synergistic effects seen in our in vitro experiments (Fig. 8a). The combination also showed an effect on the colonisation of SKOV3[WT] cells in lungs (Fig. 8b) which mirrored the antitumoral effect observed when tumour cells were implanted subcutaneously.

We also looked at the impact of the HTCT on HERV-V2 mRNA and protein levels by treating OvCar3 cells with lower drug doses compared to SKOV3[WT] and subjecting the remaining tumour mass to qPCR and IHC: HERV-V2 mRNA and protein levels mirrored the observed in vitro effects after romidepsin-vesatolimod combination treatment (Fig. 8c), while vesatolimod alone had a distinct reducing effect (Fig. 8d).

## Discussion

In this work, we have extrapolated the SaK approach to the therapy of OC, led by the observation that ovarian tumour cells express endoretroviral structural elements which can be boosted by adding HDACis (Supplementary Fig. 21). Moreover, HERVs might have the ability to activate the immune response[36]. To explore alternative approaches to antibody-mediated cell killing, we sought to recruit innate immunity mediators by agonism of TLR7/8 with vesatolimod or imiquimod. In this context, the use of TLR7/8as proved critical because, the presence of the GU-rich signature of viral RNAs[37] alone is insufficient to activate this receptor and trigger significant apoptosis in cancer cells.

Vorinostat and romidepsin are drugs already used in the treatment of both solid and haematologic malignancies[38,39], but they are also potent LRAs[40]. We found that both drugs are capable of reactivating different HERVs which are mostly epigenetically repressed. Among them, HERV-V2 which codes for a conserved envelope protein is the most prominent HERV gene with transcriptional dependency on H3 acetylation at K9.

The TLR7/8as imiquimod and vesatolimod have both antiproliferative and antiviral properties, and are already in clinical use. Both compounds show no significant cytotoxicity in the ovarian carcinoma models evaluated. By combining the two substance classes—HDACis and TLR7/8as—we were able to deliver a synergistic antiproliferative effect at subtoxic isodoses, meaning reduced side effects in a potential therapeutic setting.

Contrarily, in the evaluation of healthy PBMCs from more than 10 donors, HERV-V2 gene expression was not upregulated after HDACi exposure. Other groups working with the SaK approach in an HIV context have analysed the possible HDACi-driven systemic HERV overexpression, but found that most of the elements studied were not affected[27]. The pairing of HDACis and TLR7/8as produced a synergistic cytotoxic effect that triggered intrinsic apoptosis in OC cells. In contrast, HTCT in PBMCs derived from healthy controls showed no synergistic effect.

Our analysis of different apoptotic pathways provided clues on the mechanisms with which HTCT attacks cancer cells. Remarkably, all ovarian cancer cells evaluated underwent intrinsic apoptosis after HTCT, which was practically absent after treatment with the single drugs at subtoxic doses. The PARP enzyme was selectively cleaved as part of the HTCT-induced apoptotic process.

TLR7 gene ablation reduced apoptosis, which underscores its critical involvement in this process. In addition, ChIP studies showed that, among all HERV elements upregulated by HDACis, the transcription of HERV-V2, which exists in a single copy on chromosome 19, is predominantly increased upon HDACi exposure. The influence of HDACis on the expression of HERV-V2 is dependent on the H3Ack9, revealing some clues on the expression control of this gene. In this regard, romidepsin is more potent in modulating this effect as revealed by ChIP-seq studies.

To determine its role as a possible ligand for TLR7, we ablated HERV-V2 and observed a significant decline in apoptosis, demonstrating that this gene plays a direct role as an apoptosis mediator via TLR7. Having said this, we cannot rule out the involvement of other HERV elements which are co-expressed upon HDACi exposure. For example, the vast group of HERV-H

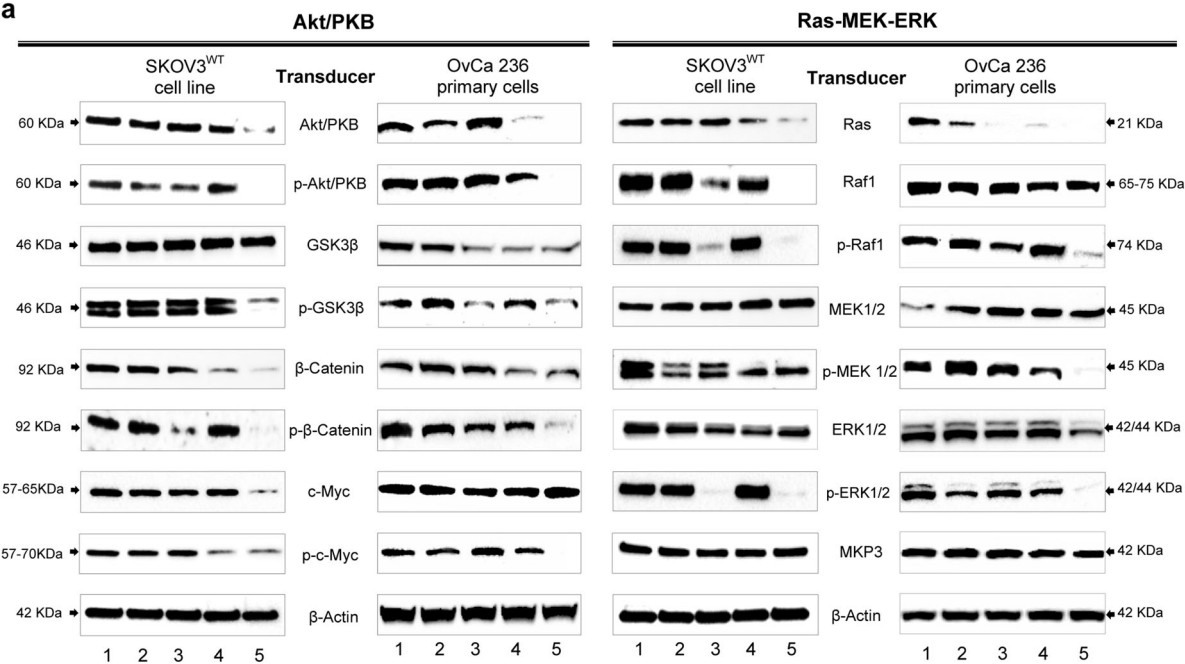

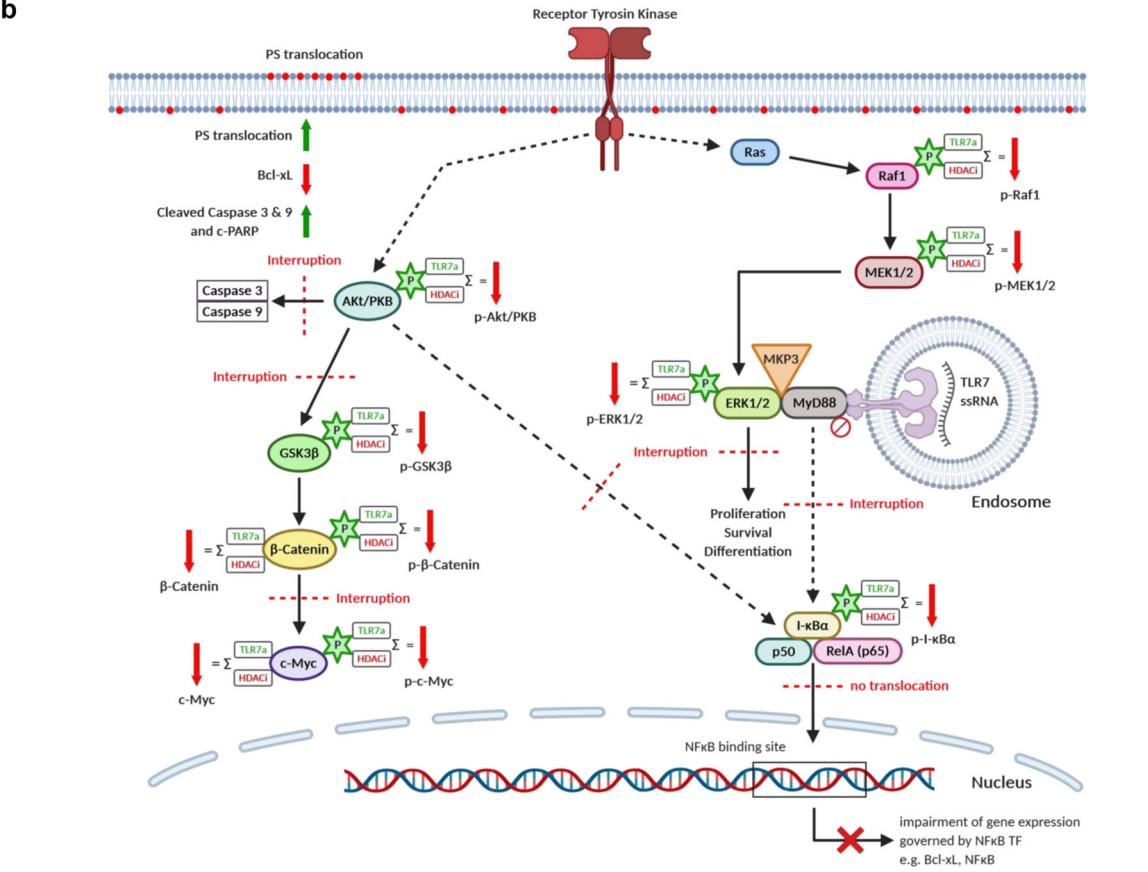

**Fig. 7 Akt/PKB and Ras-MEK-ERK signal transduction pathways are disrupted by HTCT. a** IKBα (see Fig. 6) is phosphorylated via Akt/PKB transducers, which were largely disrupted by HTCT in SKOV3^WT and OvCa236 carcinoma cells. Note the selective hypophosphorylation steady state of the MEK-ERK transducers. Protein synthesis of transducers of both cascades is affected by the exposure to single drugs and their combinations, in particular for some elements of the Akt/PKB pathway. The phosphate exchange between these transducers is globally disrupted. **b** Sketch depicting global disruption points of the TLR7-NFκB, Akt/PKB and Ras-MEK-ERK cascades for HDACis, TLR7/8as and their combinations in OC cells (Illustration created with BioRender.com). Western blot results are representative of several experiments carried out in cell lines and primary ovarian carcinoma cells isolated from ascites.

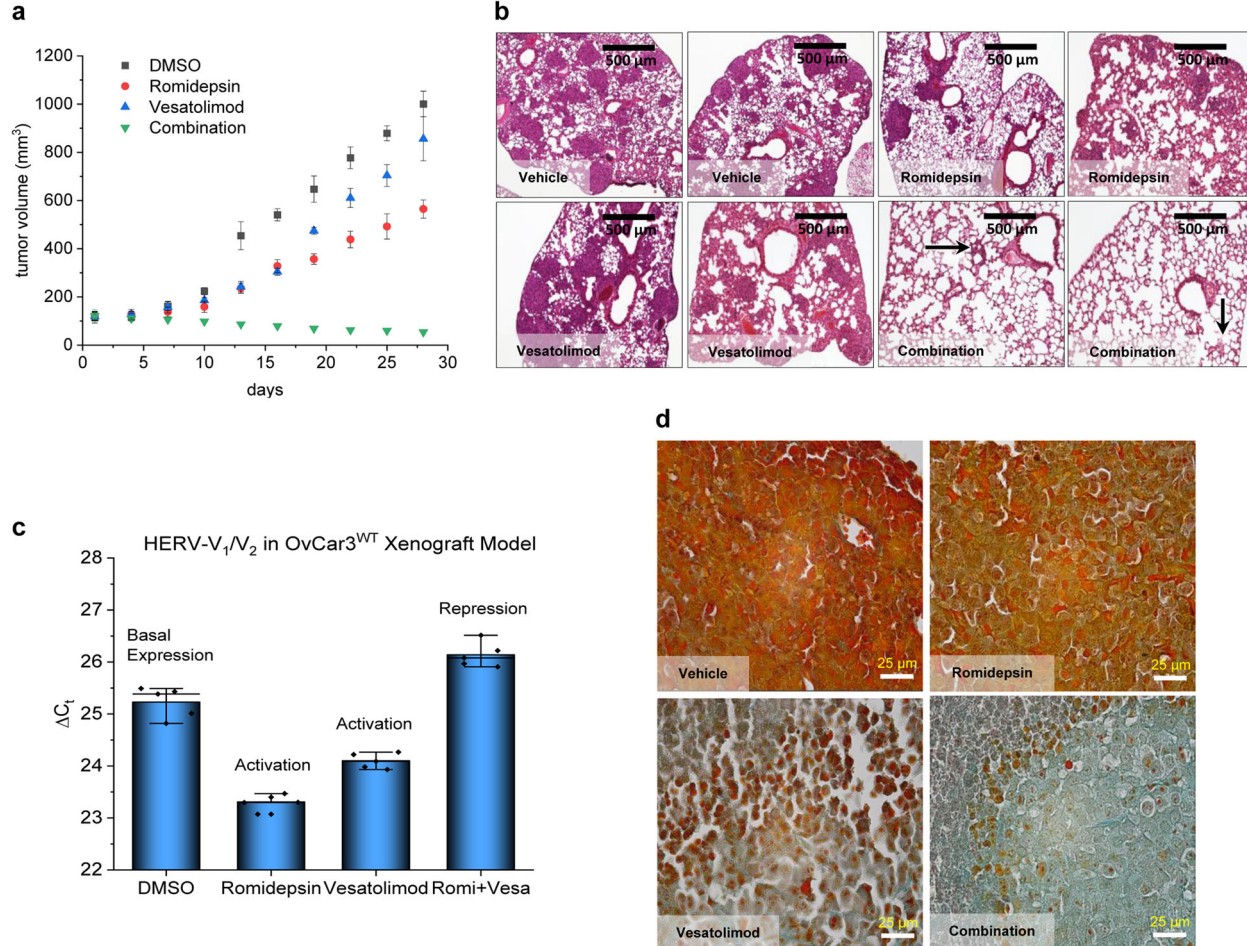

**Fig. 8 Ovarian carcinoma xenograft model in NMRI nu/nu mice. a** Tumour-bearing female NMRI nude mice were intraperitoneally treated four times at 4-day intervals with romidepsin or vesatolimod at a dose of 1 mg/kg and 5 mg/kg body weight, respectively, and with the corresponding drug combinations. Tumour volumes were calculated and plotted. Both romidepsin and vesatolimod alone show a mild antitumoral activity, as determined by the tumour size. However, when combined, their antitumoral activity is clearly enhanced. **b** In the dissemination model, mice were treated as described above after 3 days of cell inoculation, and the lungs were examined by HE staining. Under HTCT, colonisation of the lungs was significantly reduced in comparison to the single-drug treatments. **c** OvCar3$^{WT}$ xenograft model treated with subtherapeutic doses (0.5 and 2.5 mg/kg of romidepsin and vesatolimod, respectively); the remnant tumours were examined for HERV-V1/V2 expression by qPCR and returned similar results to the in vitro studies. **d** IHC of DAB staining for HERV-V1/V2 in OvCar3$^{WT}$ remnant tumours. A reduction in the expression of these retroviral proteins is observed. The graphs are representative of $n = 3$ experiments.

genes which are dispersed over several chromosomes[41]. Nevertheless, HERV-V2 is a key mediator of these effects, given that in *HERV-V2* knockout SKOV3 cell model, a significant reduction in apoptosis after HTCT is observed.

To mechanistically explain the apoptosis induced by HTCT, pivotal signalling cascades such as TLR7-NFκB, Akt/PKB and Ras-MEK-ERK were analysed. Downstream signalling between TLR7 and the transcription factor NFκB classically includes the adaptor protein MyD88. This transducer was downregulated at the protein level by HTCT at sub-cytotoxic doses. This observation raises questions about possible downstream signalling effects in the absence of this adaptor. Since SARM1 has recently been reported to act as an alternative TLR7 adaptor in neurons which also mediates apoptosis[30], we addressed this issue in OC cells in which this adaptor protein was mutated and found that SARM1 does not appear to be involved in TLR7 signal transduction.

We hence propose that, in the absence of other known adaptors, MyD88 downregulation is a secondary, indirect effect. We observed that NFκB protein is reduced after HTCT, which applies most notably to the full-length, undigested p105 form and to a lesser extent to the cleaved form p50, possibly due to a cellular pool of p50 which already existed before the treatment. This effect suggests that NFκB function was disrupted, given that it regulates its own transcription.

Akt/PKB cascade phosphorylates a plethora of proteins largely concatenated with malignant transformation and cancer cell survival[42]. We found that HTCT disrupted several transducers which belong to this cascade. Importantly, I-κBα hypophosphorylation at serine 32 was linked to the inhibition of Akt/PKB signal transduction. HTCT left I-κBα in a hypophosphorylated steady state, thus stabilising the NFκB/I-κBα complex, impeding nuclear translocation of NFκB[43], and suppressing the transcription of genes directly governed by NFκB, including Bcl-xL which was selectively downregulated by HTCT. Thus, there was not only an impairment of the active form of NFκB, but also a concurrent downregulation of other TF belonging to other cascades, e.g. c-Myc, which in turn govern a large number of genes involved in cell proliferation and the activation of apoptosis-related effectors[44].

In addition to Bcl-xL, which was used as a sentinel to monitor NFκB function, we investigated the transcriptional status of IL8, which is in part transcribed by NFκB. IL8 has been established as

a negative outcome marker in several malignant tumours, including OC[45]. Others authors found beneficial roles of IL8 in OC[46]. It is a rather promiscuous interleukin which can dock to different receptors in an autocrine or paracrine fashion, among which CXCR1 is the most prominent in ovarian carcinoma cell lines. We observed that, in various ovarian cancers including cell lines and primary ovarian carcinoma cells, HTCT resulted in IL8 downregulation, whereas the CXCR1 receptor was strongly upregulated, probably as a compensatory homoeostatic reaction. However, this effect was not observed in SKOV3[WT]. This might be explained by the possible involvement of TFs other than NFκB in IL8 expression[32].

The reduction of specific HERV elements has been reported to affect the expression of some IFN-induced genes and impair the IFN network, a key player in innate immunity which promotes the expression of IFN-stimulated genes (ISGs)[47]. Type I and II interferons and the associated ISGs are pleiotropic cytokines which help maintain the balance of immune escape and/or immune surveillance via a range of mechanisms[48]. In cancer biology, they may play a dual role in cell immunoediting. In addition to their antitumor activity via cytotoxic lymphocyte responses, they have been observed to indirectly promote tumour initiation specifically by selecting tumour cells with immunoevasive features. This has been demonstrated for IFN-ɣ, which can also help in tumour clearance[49,50].

Since HTCT, which include TLR7/8as, induces HERV-V2 downregulation, we studied the potential influence of this experimental approach on the inflammasome in regard to pathways mediated via NF-κB. Because HTCT had a predominant inhibitory effect on pro-inflammatory cytokine expression at the transcript level, the cell death signature was more consistent with intrinsic apoptosis than the inflammatory form of cell death, i.e. pyroptosis[51]. A notable exception was the HTCT-induced transcriptional upregulation of IFNɣ and of TNFα, the latter being a pleiotropic cytokine which is a key protagonist in immune homoeostasis, host defence and tumour surveillance.

In our ovarian carcinoma xenograft mouse models, the combination of romidepsin and vesatolimod showed potent antitumoral effects at subtherapeutic doses, indicating a vast therapeutic potential. A translation of our approach to cancer therapy appears clinically possible, given that most drugs analysed in this study are already approved for other indications. Still, a dose-escalation study and/or investigation of alternative routes of application in animal models are highly desirable.

To conclude, we present here a new anticancer concept of potential translational relevance. It is based on the combined use of HDACis and TLR7as to induce a synergistic cytotoxic effect in OC cells at sub-cytotoxic isodoses. This effect builds upon the reactivation of specific endogenous retroviral elements by the HDAC inhibitor and subsequent apoptosis induction by the agonist of TLR7, a key constituent of the innate immune system, which is involved in the recognition of viral signatures. Evasion of major inflammatory factors makes this therapeutic strategy attractive as a potential option for the treatment of chemoresistant cancer. Thus, we suggest that this approach should be further tested in clinical phase-I trials.

## Methods

**Ethical considerations, patient samples and cell lines**. This study was approved by the Ethics Commission of the Ruhr-University of Bochum, Medical School (register numbers: 4042-11 and 5235-15). Written informed consent was obtained from each participant. Samples were anonymized, coded and made accessible only to research staff. Primary ovarian carcinoma cells were isolated from ascites obtained by paracentesis and adapted to chemically defined culture media until growth kinetics were comparable to the cell lines employed, SKOV3[WT] (ATCC® HTB-77™) and NIH:OVCAR-3 (ATCC® HTB-161™), which were obtained from the tumour bank of the University of Duisburg-Essen, Medical School. Cells were cultured in DMEM medium containing 10% heat-inactivated foetal calf serum. All cell cultures were periodically tested for mycoplasma using MycoSPY® kit from Biontex.

**IC$_{50}$ values**. IC$_{50}$ values were determined using the MTT [3-(4,5-dimethylthiazol-2-yl)-2,5-diphenyltetrazolium bromide] proliferation assay as previously described[52]. Briefly, cells in exponential growth were trypsinised, harvested, washed with medium and seeded in 96-well plates at appropriate densities according to their growth kinetics previously established. After an adherence period of 24 h, cells were exposed to increasing concentrations of TLR7as imiquimod or vesatolimod and HDACis vorinostat or romidepsin for 72 h. The cultures were then incubated with MTT (Sigma-Aldrich) dissolved in phosphate-buffered saline (PBS) at a final concentration of 1 mg/mL for 4 h. Supernatants were aspirated, and the purple formazan crystals were dissolved in 100 μL of solubilizing solution (10% SDS in DMSO; Sigma-Aldrich). Colorimetric measurements were performed in a plate reader (Infinite F200 Tecan, Grödig, Austria) at 570 nm.

**Analysis of the interaction of HDACis and TLR7as in SKOV3$^{WT}$ cells**. The effect of both drug classes administered simultaneously was analysed by isobolography as described previously[52]. The HDACis vorinostat and romidepsin, and the TLR7as vesatolimod and imiquimod were purchased from Adooq Bioscience (CA, USA). For isodoses, the IC$_{50}$ concentrations of each substance were employed. Cells were cultured in 96-well plates to appropriate densities and allowed to adhere for 24 h. Applying fixed percentages of the IC$_{50}$ values for the first drug (10% steps from 10 to 100%) and varying the concentration of the second drug from 0.01 to 50 μg/ml, the variation in the resulting IC$_{50}$ was determined for every percentage. The same procedure was carried out inversely for the second drug. Dose–response curves were plotted. The binary interaction of the substances was evaluated in the normalized isobolograms taking into consideration in the Cartesian plane the 100–100 straight transversal line and given as additive if the resulted curves were in the proximity of this axis, antagonistic above this line and synergistic if the curves were below[53].

**qPCR analysis of the activation of HERV transcripts and inflammasome**. To reactivate latent HERV-V1/V2, SKOV3[WT] or ovarian cancer cells were exposed to 0.5×, 0.75× and 1× IC$_{50}$ of the HDACis for 24 h. In order to analyse the influence of TLR7 agonists on HERV-V1/V2 chemical activation, the HDAC inhibitors vorinostat or romidepsin and the TLR7 agonists vesatolimod or imiquimod were co-administered at 0.5×, 0.75× and 1× IC$_{50}$ for 24 h.

Following drug exposure, the cells were harvested, and RNA was extracted with Trizol® (Life Technologies, California, USA), treated with 7 Kunitz units of DNase I (Qiagen, Hilden, Germany) and further purified on RNeasy mini columns (Qiagen, Hilden, Germany) according to the manufacturer's instructions. RNA integrity was ascertained using the 2100 Bioanalyzer (Agilent, Santa Clara, CA). Four micrograms of pure and intact RNA was used for first-strand cDNA synthesis with the High Capacity cDNA RT Kit (Life Technologies, CA, USA).

For amplification and detection of HERV-V1 (NM_152473.2), HERV-V2 (NM_001191055.2) and TLR7 (NM_016562.3), specific primers and probes were designed by us (Supplementary Table 1) and synthesised by IDT Inc. (Iowa, USA). The amplification of 100 ng of cDNA was performed in triplicate with 250 nmol primers in PrimeTime® Gene Expression Master Mix (IDT, Iowa, USA) on a CFX96TM Real-Time System (Biorad Laboratories, CA, USA). Relative expression was determined using the $\Delta C_t$ comparative method.

Inflammasome mediators were analysed by qPCR in SKOV3[WT] cells after 24 h of drug exposure. RNA and cDNA were prepared as described above. Primers and probes were designed by us and synthesised by IDT Inc. (Supplementary Table 4).

**Detection of phosphatidylserine translocation as an apoptotic event upon treatment with HDACis and TLR7as**. Phosphatidylserine (PS) translocation as an early apoptosis-related event was cytometrically detected with Annexin V conjugated with Alexa Fluor 488 (San Diego, USA). Briefly, $500 \times 10^3$ cells were treated with 1× IC$_{50}$ for 24 h as described above. Cells were pelleted and resuspended in 100 μl 1× Annexin V binding buffer. 5 μl Annexin V and propidium iodide (0.1 mg/ml) for late apoptosis detection were added and incubated in the dark at room temperature for 15 min. Cells were washed, pelleted, gently mixed in 500 μl binding buffer, and measured cytometrically with a CytoFLEX Research Cytometer B5-R5-V5 (Beckman Coulter, Krefeld, Germany).

**CRISPR/Cas9 ablation of genes encoding HERV-V1 envelope protein and TLR7**. CRISPR/Cas9 (clustered regularly interspaced short palindromic repeats/CRISPR associated protein) was used to knockout *HERV-V1* and *TLR7* genes. We used the Alt-R™ CRISPR/Cas9 System and the HPRT gRNA controls from IDT Technologies, Iowa, USA, following the manufacturer's recommendations. crRNAs targeting HERV-V1/V2 and TLR7 were specifically designed using the IDT tool https://eu.idtdna.com/site/order/designtool/index/CRISPR_CUSTOM and synthesized by them. The gRNA sequences were aligned to the human genome using the Basic Local Alignment Search Tool (BLAST) to look for sites with similarity to other HERVs and tested for possible off-targets (Supplementary Fig. 4). The complex crRNA:tracrRNA was mixed at equimolar ratio in the Nuclease-Free

Duplex IDT buffer, incubated at 95 °C for 5 min and cooled down to room temperature. The ribonucleoprotein (RNP) complexes were formed according to IDT instructions by incubating 0.1 μM of the specific crRNA:tracrRNA complexes, 1 U of Alt-R® S.p. HiFi Cas9 Nuclease V3 (IDT Technologies) for 20 min at room temperature and this RNP complex was immediately transfected to SKOV3^WT using the 4D-NucleofectorTM System (Lonza) applying the pulse code FE-132.

Gene edition was corroborated with the EnGen^TM Mutation Detection Kit (New England BioLabs, MA, USA). First, genomic DNA was isolated with the NucleoSpin DNA RapidLyse Kit (Macherey-Nagel). Amplicons were obtained using endpoint PCR and the primer DNA from controls. Wild-type and mutated PCR products were subjected to heteroduplex formation (95 °C, 5′ for denaturation and 95 °C → 25 °C, at 0.2 °C/s for annealing). Next, heteroduplexes were treated with EnGen T7 endonuclease I at 37 °C for 15 min. DNA fragments were electrophoretically resolved in 2.0% agarose gel and stained with ethidium bromide. Gels were imaged using a ChemiDoc^TM XRS + Imaging System (Bio-Rad, CA, USA).

**NanoLuc® Luciferase assay for determining activation of NF-κB by different substances**. The activation of NF-κB was determined by employing the plasmids pNL3.2-NF-κB-RE [NlucP/NF-kB-RE/Hygro] and pNL2.2 [NlucP/Hygro] (Promega, WI, USA) as a negative control following the recommendation of the manufacturers. Briefly, $1 \times 10^6$ cells were mixed with 1 μg of plasmid in 100 μl SF transfection buffer (Lonza, Basel, Switzerland) and deposited in a 100 μl Nucleo-cuvette^TM. Cells were transfected with the 4D-Nucleofector^TM System (Lonza) applying the pulse code FE-132 and 3 days thereafter positive clones were selected by incubating cultures with 200 μg/ml Hygromycin B (Invitrogen/Thermofisher, CA, USA). Established cell lines were seeded $1 \times 10^4$ per well in 96-well plates suited for bioluminescence. After 24 h cells were treated with $1 \times IC_{50}$ doses of HDCAis and TLR7as. Again, 24 h later, cells were incubated with TNF-α (Biolegend, CA, USA) 5 h prior to measurement. The occurring bioluminescence was detected by using Nano-Glo® Luciferase Assay System (Promega, WI, USA) and a plate reader (Infinite F200 Tecan, Grödig, Austria).

**ChIP assay**. Chromatin immunoprecipitation[54] was performed using SimpleChIP® Enzymatic Chromatin IP Kit (Magnetic Beads) #9003 from Cell Signaling. After treating cells with HDCAis for 24 h, proteins were cross-linked to DNA by adding formaldehyde (1% v/v final concentration) to culture media followed by an incubation of the cells for 10 min at room temperature (RT). To stop the crosslinking process glycine (2% v/v final concentration) was added and cells were incubated for 5 min at RT. Cells were washed twice and their nuclei extracted after incubating the cells in 1× Buffer A + DTT + PIC (protease inhibitor cocktail) on ice for 10 min and several centrifugation steps. Micrococcal nuclease was added to the tube and incubated for 20 min at 37 °C with frequent mixing to digest DNA to a length of ~150–900 bp. Digestion was stopped by adding EDTA (0.05 M final concentration) and the tube was placed on ice for 2 min. After pelleting and resuspending in 100 μl of 1× ChIP Buffer + PIC Nuclei were lysed by sonicating the sample for 20 s, three times by incubating for 30 s on wet ice in between pulses. Lysate was clarified by centrifuging at 9400 RCF for 10 min at 4 °C and the supernatant was transferred to a new tube. A small portion of this cross-linked chromatin was analysed for digestion and concentration. From the cross-linked chromatin, 20 μg was taken for each IP. This 20 μg chromatin was made up to 500 μl with 1× Chip Buffer + PIC and 2% (10 μl) of this diluted chromatin was set aside as input sample and stored at −20 °C until further use. To this diluted chromatin, immunoprecipitating antibody, spike in chromatin and spike in antibody were added according to manufacturer's recommendation. As reference genome in the spike in control Dm3 Drosophila melanogaster (active motif, CA, USA) was used. IP samples were incubated overnight at 4 °C with rotation. After incubation, ChIP-Grade Protein G Magnetic Beads were added to each IP reaction and incubated for 2 h at 4 °C with rotation. Protein G magnetic beads were pelleted using a magnetic separation rack, the supernatant was removed and the beads were resuspended in low salt wash buffer and incubated at 4 °C for 5 min with rotation. Beads were pelleted and supernatant was discarded after incubation. This step was performed thrice. High salt wash buffer was added to the beads and incubated at 4 °C for 5 min with rotation. After this step, beads were pelleted and supernatant was discarded. 1× ChIP Elution Buffer was added to the 2% input sample tube and set aside at room temperature. 1× Chip Elution Buffer was added to the beads and chromatin was eluted for 30 min at 65 °C at 1200 rpm using a thermomixer. Beads were pelleted using magnetic separation rack and eluted chromatin supernatant was carefully transferred to a new tube. To all tubes, including the 2% input sample, cross-links were reversed by adding 5 M NaCl and proteinase, and incubating for 2 h at 65 °C. After this step, chromatin is eluted using the DNA elution columns provided and ChIP sequencing was performed by BGI Genomics using Illumina next-generation sequencing (NGS) system (Hong Kong, China). The genomic positions of all peaks identified before and after the treatment of SKOV3^WT cells with either vorinostat or romidepsin have been intersected with the proviral coordinates of a collection of about 3280 individual HERV loci integrated in human genome, which have been characterised in detail in previous works[14,55–57], through bedtools "intersect" option. The same method has been used to individuate differential peaks (diffpeaks) co-localised with HERV sequences. In addition, to ChIP-seq, the purified chromatin was used for RT qPCR. The amplification of 5 ng of chromatin was

performed in triplicate on a CFX96TM Real-Time System (Biorad Laboratories, CA, USA). Relative expression was determined using the $\Delta\Delta C_t$ comparative method.

**Immunotechniques**. Immunocytochemistry, immunohistochemistry, cytometry and western blot experiments were performed using standard protocols as previously described[52].

*Western blots*. For western blot analysis, $5 \times 10^6$ cells in exponential growth cultured in 75-cm$^2$ flasks were exposed to 0.5-fold their IC$_{50}$-values for 24 h. After drug exposure, cells were washed with cold PBS, trypsinised, resuspended in DMEM containing 10% FCS and centrifuged at 300 RCF for 5 min; pellets were washed with cold PBS, centrifuged and lysed in RIPA buffer [150 mM NaCl, 1 mM EDTA, 1% Triton X-100, 1% sodium deoxycholate, 0.1% SDS, and 50 mM Tris-HCl pH 7.4] in the presence of a proteinase inhibitor cocktail according to the manufacturer's instructions (Roche Diagnostics GmbH) for 30 min on ice and then centrifuged for 20 min at 14,000 RCF (Relative Centrifuge Force) at 4 °C for 20 min. Supernatants were collected and protein concentration was measured by Bradford. Thirty micrograms of total protein were resolved by SDS-PAGE in a 4–12% gradient gel (Bio-Rad Laboratories, CA, USA) using Tris-glycine (0.025 M Tris-HCl, 0.192 M glycine pH 8.5) buffer, and transferred to 0.2 mm nitrocellulose membrane (Pierce Protein; Thermo Scientific, Inc.). Blots were blocked with 5% BSA or nonfat milk resuspended in PBST (0.05% Tween-20 in 1× PBS). Blots were washed with PBST and incubated overnight with primary antibodies resuspended in 1% BSA in PBST taking into consideration the recommendations of the manufacturers. After incubation, blots were washed 3× with PBST solution and incubated with secondary antibody conjugated with HRP resuspended in 1% BSA in PBST taking into consideration the recommendations of the manufacturers for 2 h at room temperature. Immunoblots were washed 3× with PBST and developed by Western Lightning Plus-ECL (Perkin Elmer) using a ChemiDoc XRS + system with Image Lab Version 2.0.1 software (Bio-Rad Laboratories).

*Immunocytochemistry*. Cells were grown in chamber slides to appropriate densities, washed with 1× PBS, fixed with 4% formaldehyde in PBS for 20 min, rinsed twice with 1× PBS for 5 min, and blocked with 10% normal goat serum (AbD Serotec, London, UK) in PBST at room temperature for 60 min. Wells were incubated with primary antibodies resuspended in 1% goat serum in PBST at concentrations following the manufactures' recommendations at 4 °C overnight. Chambers were washed 3× with PBST and finally incubated with the appropriate secondary antibodies resuspended in 1% goat serum in PBST at room temperature for 2 h under exclusion of light. Wells were rinsed 3× with PBST and mounted using Faramound Mounting medium (Dako) for visualization in an Eclipse i-50 microscope (Nikon), using the NIS advance imaging software.

*Immunohistochemistry*. For IHC, tissue samples were fixed with 4% formaldehyde dissolved in PBS, dehydrated and embedded in paraffin overnight. Paraffin tissue sections of 4-μm thickness were baked overnight at 60 °C to firmly attach the sections to the slides. After baking, the sections were deparaffinized in two changes of xylene-substitute (Thermo Scientific, London, UK) solution for 10–15 min and rehydrated in a series of graded ethanol solutions (100%, 100%, 95%, 70%, 50%) for 3 min each. Samples were subjected to antigen retrieval by heating the sections for 30 min in 10 mM sodium citrate buffer pH 9.0 at 95 °C in a domestic vegetable steamer. The slides were washed twice in 1× PBS for 5 min and blocked for 60 min with 10% normal goat serum at room temperature. Primary antibodies were applied overnight at 4 °C according to the manufacturers' recommendations by diluting in 1% goat serum in PBST. On the next day, the slides were washed three times in PBST for 5 min each. Tissue sections were incubated with conjugated secondary antibodies (Cell Signaling, Cambridge, UK) diluted in 2.5% goat serum in PBST for 120 min at room temperature according to the manufacturers' recommendations. Next, the samples were stained for 15 min with 1 μg/ml Hoechst 33258 diluted in PBST in order to visualize the nuclei. The slides were then washed three times in PBST for 5 min each and rinsed in 1× PBS for another 5 min. Tissue specimens were mounted using Faramound Mounting medium (Dako) for visualization in an Eclipse i-50 microscope (Nikon).

*DAB staining*. For histochemical analysis the DAB staining kit (Abcam 64238) was employed. DAB (3,3′-Diaminobenzidine) is suitable for immunohistochemical staining as it produces a heat-resistant brown chromogen which is insoluble in water, alcohol and other solvents in the presence of HRP. Briefly, after antibody retrieval, sections were washed three times with washing buffer and covered with peroxide blocking solution for 10 min following the instructions of the kit. Sections were rinsed in washing buffer and incubated with DAB substrate until a brown colour was achieved. Slides were washed with tap water ten times and covered with mounting medium.

*FACS analysis*. For cytometric analysis, SKOV3 cells were harvested and fixed with methanol/acetone (1:1) for 15 min at 4 °C and blocked with 5% BSA in PBST for 30 min at room temperature. Fixed cells were centrifuged at 500 RCF for 5 min and pellets were rinsed three times with PBST and incubated with primary antibodies

diluted in 1% BSA in PBST for 2 h at room temperature according to manufacturer's recommendations. Cells were washed three times with PBST, and incubated with secondary antibodies diluted at 1:1000 in 1% BSA in PBST for 1 h. Conjugated antibodies were employed according to manufacturer's recommendations. Appropriate isotype antibodies were used for cytometric analysis. Labelled cells were cytometrically measured using a CytoFLEX Research Cytometer B5-R5-V5 (Beckman Coulter, Krefeld, Germany). The main population of the cells was gated according to the FSC/SSC dot plot density, discriminating them from the dead cells. Signals of isotype controls were used to set the gate for the negative signals.

**Tumour xenograft model**. The effectiveness of the HDACi/TLR7a combination therapy was determined in vivo in a SKOV3$^{WT}$ xenograft model. $1 \times 10^7$ SKOV3$^{WT}$ cells obtained from 2D/3D cultures were subcutaneously injected into the right upper flanks of 6- to 8-week-old female NMRI nude mice. One week after the tumours were measurable, the mice were randomized into groups of four animals each. Romidepsin and vesatolimod were intraperitoneally administered four times at 4-day intervals at doses of 1 and 5 mg/kg body weight, respectively. Tumours were measured using a caliper and their respective volumes calculated as $V = 0.52 \times \text{length} \times \text{width}^2$. At day 28, the animals were killed by cervical dislocation after isoflurane narcotization. For tumour dissemination studies $1 \times 10^6$ of SKOV3$^{WT}$ cells were i.v. injected into female NMRI nude mice, via the tail vein. Treatments were performed as above described, 3 days after tumour cell inoculation. At day 14, animals were narcotized in an isoflurane atmosphere and sacrificed by cervical dislocation. Lungs were extracted and fixed by immersion in 4% formaldehyde overnight and then stored in 70% ethanol prior to paraffin embedding. H&E staining was performed as previously described[52,58]. The study was performed in accordance with the Animal Welfare Act and was approved by the NRW law under the register number AZ81-02.04.2019.A334.

**Statistics and reproducibility**. Intergroup comparisons of medians were performed with paired Student's $t$ test, two-tailed. Significance was defined as $p < 0.05$; statistical analysis was done with OriginLab2021 (OriginLab Corp., MA, USA). Significance was accepted when $p < 0.05$. Western blots and immunochemistry studies were descriptive and therefore not analysed statistically. All results shown are representative of at least three independent experiments.

**Reporting summary**. Further information on research design is available in the Nature Research Reporting Summary linked to this article.

## Data availability

ChIP-seq data from this study was deposited in NCBI GEO under the accession number GSE164809. Raw data corresponding to all the main figures is provided as Supplementary Data 2 file linked to this article. All the antibodies used in this study and other information regarding experimental design can be found in reporting summary. All other data are available from the corresponding author on reasonable request.

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

## Acknowledgements
The authors want to thank Marienhospital Herne, Elisabeth Group, for supporting this study. They would like to extend special thanks to the staff of the Gynaecology Department for the coordination of sample collection. We regret if we failed to mention relevant studies in the field published by others.

## Author contributions
Conception and design: D.D.-C. and S.S. Development of methodology: D.D.-C. and S.S. Acquisition of data: D.D.-C., S.S, A.H.A, E.H., J.E., R.D., S.A., A.N.-B., E.P., S.M., F.D.'S., J.K., P.D., D.K., A.N., M.T., K.S., N.G. and C.O. Analysis and interpretation of data: D.D.-C., S.S., D.S., E.T. and N.G. Writing, review and/or revision of the manuscript: D.D.-C., S. Saka, S. Malak and M. Teipel. Administrative, technical, or material support: D.D.-C., D.K., A.T. and D.S. Study supervision: D.D.-C., S.S. and D.S.

## Funding

## Competing interests
The authors declare no competing interests.
