## [Peer Review File · Communications Biology]

Reviewers' comments:

Reviewer #1 (Remarks to the Author):

In this paper, Díaz-Carballo et al. examine the regulation of endogenous retroviral RNA transcription by HDAC inhibition and suggest that coupling this with RNA sensing via TLR7 agonism can lead to increased intrinsic tumor cell death. While the de-repression of endogenous retroviral elements by epigenetic regulation is a novel and interesting idea, limitations in experimental design and interpretation hinder the conclusion that the observed cell death in tumor exposed to HDAC inhibitors and TLR7 agonists is due to this mechanism. Overall, confirmatory experiments using the HERV-V1/V2 tumor cell line are needed to ensure that the effects on cell death can be attributed to de-repression of these endogenous retroviral RNAs.

Major critiques:

1) While the data concerning the transcriptional upregulation of HERV-V1/V2 upon HDAC inhibition is convincing, it is unclear whether this is a direct effect of the drugs on epigenetic marks proximate to sites of HERV-V1/V2 integration. ChIP-seq analysis of relevant histone(s) would help resolve this question. Also, it would be surprising if a single HERV subclass dominates the effect. It would be interesting/important to see where HERV-V1/V2 ranks among all induced HERVs in vorinostat treated SKOV3 cells.

2) The presentation of the qPCR data in Fig. 2 is confusing. The presentation of the amplification curves and deltaCt values is unnecessary and all relative quantification should be done using the same method (e.g. deltaCt as shown in Fig. S2a. Indeed, the data presented in Fig. S2a is convincing and should take the place of the current Fig. 2a). As presented, it is unclear whether the data in Fig. 2c supports the authors conclusions.

Also, later in the text, it is stated that "The combination of Romidepsin and Vesatolimod triggered apoptosis in 25% of the treated wildtype cells, which correlated with the reduction in RNA transcripts as seen in our qPCR analysis." RT-qPCR is usually assumed to measure the transcripts present in live cells, not as a method of quantifying loss of transcripts in dying cells. As presented, the consensus interpretation of Fig. 2a would be that the cells remaining after combination therapy downregulated the HERV-V1 transcript (relative to vorinostat alone). This statement calls into question the overall interpretation of the qPCR results in the paper.

3) The presentation of the flow cytometry data in Fig. 2b and Fig. S2 should be done in a way that allows direct comparison (e.g. staggered overlay), and needs a figure legend. Also, if the red line indicated untreated/mock treated cells, an isotype control additionally displayed allow determination of basal expression. Additionally, the reason for the discordance between the flow cytometry and qPCR results is unclear, and should be addressed in the text. Lastly, it is unclear whether diminution of expression in co-treated samples could only reflect cell death rather than true decrease of protein expression. Dead cell exclusion should be performed.

4) The statement: "...the combination of HDACis and TLR7as did not trigger apoptosis in PBMCs derived from healthy individuals (S4)" does not appear to accurate. While apoptosis did occur, there may not be a synergistic effect with the combination of HDACi and TLR7a, though without error bars and statistical analysis, this is difficult to discern.

5) It is unclear whether the experiment in Fig. 4a is confounded by the presence of exogenous RNA species as the scrambled siRNA does not appear to be shown (unless this is "- siRNA", which if true, an untransduced control should be shown for comparison). Regardless, this data should be excised as

it is non-contributory. If exogenous RNA is confounding, could this also apply to the CRISPR-Cas9 system, where guide RNAs are produced? Additionally, lack of derepression of HERV-V1/V2 after HDACi should be confirmed in the KO cell line.

6) The results of inflammatory signaling in Fig. 5 under HDACi and TLR7a are unclear. Given that MYD88 is the adaptor for TLR7, should it not increase with TLR7a (for instance as seen here with TLR4 agonism: DOI: 10.4049/jimmunol.1203441)? How do the authors conceptually explain the dramatic downregulation of MYD88 and NFkB under combination treatment in Fig. 5b?

7) Figure 6b is overly complicated and contains claims not addressed in the paper. It should limit its scope to the data presented within the paper or eliminated.

8) It is not clear that the in vivo diminution of tumor growth is due to the de-repression of HERV-V transcripts coupled with TLR7a rather than a direct cytotoxic effect mediated by other factors. Conducting the experiment with the HERV-V1/V2 CRISPR KO cell line would add credence to this hypothesis.

Minor critiques:

1) The claims in paragraph 5 of the Introduction should be referenced.

2) The figure legend for Fig. S1c needs correction.

3) Figure S2b-c are not referenced in the text and the results are unconvincing (S2b: 96 vs. 92% as upregulation, and does not follow dose-response curve). Further, Fig. S2c appears mislabeled as Fig. S3c.

4) The labels for the flow cytometry in Fig. 3 should be made more legible.

5) Fig. 4b-i, should have a ladder and the expected size of the product should be indicated. Also, are multiple HERV-V1/V2 integration sites located across the genome? Does CRISPR/Cas9 silencing account for all of them?

6) Higher power insets in Fig. 7b would be helpful to visualize tumor cells.

7) Statistical analyses should be clearly described in all figure legends. Additionally, if statistical significance is claimed, error bars should be shown.

8) The language used, at times, is somewhat colloquial (e.g. cells "commit suicide"), and there are translation errors in the text (e.g. mamma instead of breast). Please proofread.

Reviewer #2 (Remarks to the Author):

In the presented study the authors investigate reactivation of human endogenous retroviral (HERV) elements via histone deacetylase inhibitors (HDACi) as a potential therapeutic approach in cancers. In detail, the authors report that HDACi treatment sensitizes cancer cells to apoptosis induction via interferon response pathways responding to re-expressed HERV RNAs, and that this pathway can be targeted with combination of HDACi and TLR7 agonists like imiquimod. The study illustrates the effectiveness of this approach in vitro, as well as in vivo xenograft models. The possible mechanism of action for the HDACi/TLR7 agonist combination is investigated and revealed induction of apoptosis, together with signaling pathway alterations as possible causes for the synergistic drug effects. The manuscript is structured very clearly and data are presented in a logical, transparent way. Strengths include data are corroborated by using different drugs/drug combinations, different drug concentrations and in some instances multiple different cell models. The presented conclusions are of value to our understanding of how immune-stimulatory drugs like imiquimod may be used, and have important translational implications. A major weakness is the study strongly relies on the SKOV3 cell line as a model to draw its conclusions. While some experiments (e.g. CRISPR/Cas9 knockout) may be

difficult to perform in primary cells, other experiments, such as HERV-V1/V2 expression and downstream effects on signaling and apoptosis pathways, would be feasible in primary cells.

In the results section “synergistic cytotoxic effects of HDACis and TLR7as in SKOV3WT and primary ovarian cancer cells”, the study mentions the use of three primary human ovarian cancer cell isolates (OvCa28, OvCa29 and OvCa236). However, data in Fig 1 and Fig. S1 only show drug toxicity for two of these (OvCa28 and OvCa236), and isobolograms are only shown for OvCa236. In order to show the reproducibility of their findings, the authors have to present cytotoxicity and isobologram data for all three ovarian cancer isolates, and may comment on any observed differences between the three. In the results section “Simultaneous exposure to HDACis and TLR7as interferes with transcriptional expression of HERV-V1 but not of TLR7”, the findings presented for HERV-V1/V2 RNA expression are key for most of the study’s conclusions, but are only being shown for the SKOV3 cell line. In order to justify the conclusions made by the authors regarding the translational significance of the study, the HERV expression under drug treatment needs to be assessed in primary cell samples in vitro. This would ideally be done in the three ovarian cancer isolates (OvCa28, OvCa29 and OvCa236) already presented in Fig. 1 and Fig S1.

In the results sections “HDACi-TLR7a combination treatment is a potent, selective inducer of intrinsic apoptosis, Effects of HDACis and TLR7as on inflammatory factors and Combination of HDACis and TLR7as impairs signal transduction pathways”, these three paragraphs highlight potential mechanisms leading to the observed drug combination effects, namely induction of apoptosis and alterations in signaling pathways like NFKB, MEK/ERK and AKT. Alterations were shown to take place on the transcriptional (Fig. 5A) and translational/post-translational level (Fig. 5B, Fig. 6). However, the authors again only show data for the SKOV3 cell line, leading to the question whether any of the observed mechanisms could be recapitulated in other cell lines or primary samples. The authors need to address this question by repeating some of their mechanistic experiments in other cell models, preferentially primary cell isolates like the ones shown already in Fig. 1.

In the results section “HERV-V1/V2 gene ablation with CRISPR/Cas9 significantly reduces apoptosis”, the presented data indicate successful gene editing with CRISPR/cas9 on the DNA level, as well as alterations in downstream effects (apoptosis, PARP cleavage), but do not confirm a change in HERV-V1/V2 RNA expression as a result of CRISPR/Cas9 treatment. HERVs are multi-copy loci and can have complex deletions, mutations and varying sequence homology. It can be expected that gene editing affects multiple loci, but leaves others intact, since editing efficiency is limited. Thus, confirmation of actual expression differences caused by CRISPR/Cas9 is essential. To justify their conclusions regarding the link between HERV transcript expression and downstream effects after drug treatment, the authors need to document alterations in HERV-V1/V2 RNA transcription in the CRISPR/Cas9-modified SKOV3 cells.

Minor points:

The drug concentrations throughout the manuscript are given in $\mu\text{g/ml}$, which makes it hard for readers to compare the presented data with other studies that use molar concentrations. The authors should – at least for their most important experiments – present drug concentrations in molar units (e.g. μM , nM) as well.

Fig. 7B seems to be missing a description in the figure legend. This needs to be added, including an explanation of what the arrows in the presented images mean.

Reviewer #3 (Remarks to the Author):

The manuscript reports a synergistic cytotoxic effect of HDAC inhibitors (HDI) and TLR7 agonists in ovarian cancer cells and xenografts. The authors also address the potential underlying mechanisms responsible for the increased apoptosis in cells treated with the combination of HDI and TLR7 agonists. Although the findings are novel and potentially significant, there are specific points summarized below that should be addressed.

1. The conclusions about NFkB are not supported by the results. Western analysis of NFkB subunits does not necessarily reflect the extent of NFkB activation. In order to make conclusions about NFkB activation, the NFkB activity should be analyzed by ChIP or a luciferase assay. In addition, it is not clear what NFkB subunit was analyzed; p65?
2. HDI have been previously reported to induce expression of the pro-inflammatory chemokine IL-8 and other NFkB-p65 dependent genes in ovarian cancer cells by increasing the activity of Ikb kinase. It would be interesting to determine whether the TLR7 agonists suppress the HDI-induced IL-8 expression in OC cells, since IL-8 has a tumor-promoting effect in OC.
3. Materials and Methods: Western blotting and the source of the antibodies used should be described.

Minor point: English should be corrected (e.g., First page, Financial support: "... grants awarded by ..."; Abstract, last line "...promising value for cancer treatments."

All changes in the figures in the manuscript are found in a tabular form at the end of this document.

Answer to the Reviewer 1

We really are very grateful for all observations and recommendations provided by this reviewer. All his/her critiques were deeply addressed in the new version.

“Major critiques:

“1) While the data concerning the transcriptional upregulation of HERV-V₁/V₂ upon HDAC inhibition is convincing, it is unclear whether this is a direct effect of the drugs on epigenetic marks proximate to sites of HERV-V₁/V₂ integration. ChIP-seq analysis of relevant histone(s) would help resolve this question. Also, it would be surprising if a single HERV subclass dominates the effect. It would be interesting/important to see where HERV-V₁/V₂ ranks among all induced HERVs in vorinostat treated SKOV3 cells.”

Response: This was a very helpful recommendation for which we are very thankful. To address this point, we performed ChIP-seq analysis of the effect of romidepsin and vorinostat in SKOV3^{WT} cell line. Moreover, we confirmed the results by ChIP-qPCR in three different ovarian carcinoma cells. The ChIP-seq analysis was performed by the group of Prof. Tramontano which has a vast experience in HERVs at genetic level, gathering an exhaustive data of around 3200 HERV loci. They found that in fact among the overall 82 HERV loci co-localized with diff peaks, only 11 had increased acetylation (and hence transcriptional activity) after HDACis treatments, and among them HERV-V₂ is the most significant in both treatments.

Results regarding ChIP-seq and ChIP-qPCR can be found in **Figure 5b, c** and **S5c, d**.

“2) The presentation of the qPCR data in Fig. 2 is confusing. The presentation of the amplification curves and deltaCt values is unnecessary and all relative quantification should be done using the same method (e.g. deltaCt as shown in Fig. S2a. Indeed, the data presented in Fig. S2a is convincing and should take the place of the current Fig. 2a). As presented, is it unclear whether the data in Fig. 2c supports the authors’ conclusions.”

Response: In the present version, we have reorganised the information from the original version to make the manuscript more fluid and better structured. The relative transcript quantification was done using the ΔC_t -values or $-\Delta\Delta C_t$ -values for facilitating the analysis of the results. The former **FigS2a** is in the present version the **Figure 5a** as recommended. The former **Figure 2c** depicted the influence of our combination in the expression of TLR7. Given that many of cytostatics negatively influence several proteins, we performed a study on how the TLR7 is affected after the therapy, if there is any negative regulation, which might represent a negative biology of one functioning biological system like the body is. Well, we want to show to the readers that this is not the case that TLR7 is not downregulated by the drugs employed and their combinations. Possibly it was not properly explained by us in the old version. In the new manuscript, it is depicted as **Figure 6b**.

“Also, later in the text, it is stated that “The combination of Romidepsin and Vesatolimod triggered apoptosis in 25% of the treated wildtype cells, which correlated with the reduction in RNA transcripts

as seen in our qPCR analysis.” RT-qPCR is usually assumed to measure the transcripts present in live cells, not as a method of quantifying loss of transcripts in dying cells. As presented, the consensus interpretation of Fig. 2a would be that the cells remaining after combination therapy downregulated the HERV-V1 transcript (relative to vorinostat alone). This statement calls into question the overall interpretation of the qPCR results in the paper.”

Response: We modified the statement since it was in fact a confounding one. Therefore, as suggested by this reviewer, we introduced the statement that “HERV-V₂ transcripts are down-regulated in the remaining cells”.

It is curious where the effects are expected to be observed in cells treated with cytostatics. We in fact, addressed the issue of the populations which remained attached to the culture surfaces or the floating cells and the expression of proteins (**Fig. S3d** in the new Supplemental Section). The populations that we analyzed in the whole study represent the remaining attached populations. We selectively discarded the floating/dying populations although they are dying by the effect of the drugs, whereas in the attached cells those processes are still occurring.

“3) The presentation of the flow cytometry data in Fig. 2b and Fig. S2 should be done in a way that allows direct comparison (e.g staggered overlay), and needs a figure legend. Also, if the red line indicated untreated/mock treated cells, an isotype control additionally displayed allow determination of basal expression. Additionally, the reason for the discordance between the flow cytometry and qPCR results is unclear, and should be addressed in the text. Lastly, it is unclear whether diminution of expression in co-treated samples could only reflect cell death rather than true decrease of protein expression. Dead cell exclusion should be performed.”

Response: The flow cytometric analyses were composed as overlaid histograms including isotype controls. We indicated this in the legends of all FACS figures.

The discordance between FACS and qPCR was addressed in the text. From our studies, it is clear that the amounts of HERV-V₁/V₂ protein in the cells are really high and cells do not synthesise more protein in spite of HDACi treatment indicating that the protein amounts and transcripts is uncoupled. The group of Chiappinelli et.al., (Cell. doi:10.1016/j.cell.2015.07.011.) also found this phenomenon after Aza treatments (Figure 6 in their paper). Nevertheless, the reduction of HERV-V₁/V₂ proteins after the combinations of HDACis and TLR7/8as is observed.

All experiments were performed using the attached populations remained after drug insults. Nevertheless, in our experience with cell cycle analysis, we were familiar since the beginning that the “Sub-G1 population” represents the dying cells. Therefore, in the cytometric measurements of whole populations, those cells were not taken into consideration for analysis. Hence, we gated only the main population supposed to be different to the dying one, which is 7ADD negative. The diminution of the proteins evaluated reflects the decrease of expression in the remaining cells. **Figure 2b** from old manuscript is now new **S3f**.

“4) The statement: “...the combination of HDACis and TLR7as did not trigger apoptosis in PBMCs derived from healthy individuals (S4)” does not appear to accurate. While apoptosis did occur, there

may not be a synergistic effect with the combination of HDACi and TLR7a, though without error bars and statistical analysis, this is difficult to discern.”

Response: Thanks for this observation. In fact, the statement in the original manuscript was not accurate. We are now presenting all results with the corresponding error bars and the proper descriptive statistics. Actually, a synergistic effect is not observed with the combination of HDACis and TLR7as and the apoptosis observed is almost same as for single drugs in case of PBMCs. This is now **S3e**.

“5) It is unclear whether the experiment in Fig. 4a is confounded by the presence of exogenous RNA species as the scrambled siRNA does not appear to be shown (unless this is “– siRNA”, which if true, an untransduced control should be shown for comparison). Regardless, this data should be excised as it is non-contributory. If exogenous RNA is confounding, could this also apply to the CRISPR-Cas9 system, where guide RNAs are produced? Additionally, lack of de-repression of HERV-V1/V2 after HDACi should be confirmed in the KO cell line.”

Response: The siRNA-data was excised from the article as proposed by this reviewer. In the original works we included validated scramble siRNA and controls purchased from Thermo Fisher Scientific. We introduced those data at that time because we wanted to report this effect. The idea of this effect to be applicable for gRNA is reasonable, and we asked ourselves the same question. We supposed at that time that because of the use of ribonucleoprotein complexes (gRNA + Cas9 with 3NLS), the recognition of those RNAs (complexed with Cas9 enzymes) by cell systems were somehow protected.

The lack of de-repression of HERV-V₁/V₂ in KO cells as effective as wild type cells was confirmed. The reason of slight de-repression in KO cells could be due to the presence of mixed population as we did not select the KO cells. Those results are gathered in the **S4g** in new supplemental information section.

“6) The results of inflammatory signaling in Fig. 5 under HDACi and TLR7a are unclear. Given that MYD88 is the adaptor for TLR7, should it not increase with TLR7a (for instance as seen here with TLR4 agonism: DOI: 10.4049/jimmunol.1203441)? How do the authors conceptually explain the dramatic downregulation of MYD88 and NFκB under combination treatment in Fig. 5b?”

Response: This topic was addressed in our present version since it is relevant. Firstly, the downregulation of MyD88 at protein levels observed is an interesting effect. It could represent a secondary terminal effect chronologically occurred as consequence of drug effects. This is now addressed in discussion. It looks that the transcription of this adapter is slightly upregulated, perhaps as compensation of the lack of MyD88 protein. All these results are now in **Figure 6**.

In case of NFκB, we could reconstruct a possible mechanism and is discussed in new version of manuscript. This was also performed in accordance with other reviewer recommendation. As result of the combination of HDACis and TLR7/8as the downregulation of this transcription factor was evident. This could be because Akt/PKB is responsible for the phosphorylation of I-κBα. Since Akt/PKB signaling is interrupted, I-κBα cannot get dissociated from the complex

p50/RelA (NFκB1) and consequently a functional NFκB1 cannot translocate to the nucleus, where it should transcribe itself. This functional impairment was deduced from the luciferase assay. These results can be found now in **Figure 6**.

“7) Figure 6b is overly complicated and contains claims not addressed in the paper. It should limit its scope to the data presented within the paper or eliminated.”

Response: We limited the items to those which were investigated, as proposed by this reviewer. This is now **Figure 7b**.

“8) It is not clear that the *in vivo* diminution of tumor growth is due to the de-repression of HERV-V transcripts coupled with TLR7a rather than a direct cytotoxic effect mediated by other factors. Conducting the experiment with the HERV-V1/V2 CRISPR KO cell line would add credence to this hypothesis.”

Response: To describe the involvement of HERV-V₁/V₂ in apoptosis, we provided evidences that apoptosis in HERV-V₁/V₂ wildtype cells was significantly higher compared to KO cells. This was addressed by epifluorescence detecting cPARP, Western blots as well as morphologically (now **Figure 4c-e** and **S4h**). The *in vivo* experiments clearly show a synergistic effect of this drug combination and the activity of single drugs was more less as in the combination. In the new **Figure 8 c & b**, we are presenting new results on what happens at RNA and protein levels in OvCar3 treated at low doses of single compounds as well as their combinations. We in fact detected a reduction on RNA levels in a similar way as presented in the old **Figure 2A**. In the same sense, we analysed HERV-V₁/V₂ at protein levels in the remnant tumour tissues and detected a reduction of protein in the combination treatment.

We in fact, planed and performed the mice experiments in NMRI nu/nu mice as this reviewer proposed, because we believe that such experiments can support the data observed. Unfortunately, the mice did not develop tumors. This was a mystery which surprised us. We cannot discern what happened, if the mice were more resistant to this tumor cell subtype or if the tumor biology after the genetic manipulation rendered in non-tumorigenic cells. From 50 mice only 3 developed a pauper tumor. Thus, another way to clear this problem somehow is using SCID mice, but this is technically not possible in the Covid19-pandemia time.

Minor critiques:

1) The claims in paragraph 5 of the Introduction should be referenced.

Response: Done, thank you.

2) The figure legend for Fig. S1c needs correction.

Response: Done, thank you.

3) Figure S2b-c are not referenced in the text and the results are unconvincing (S2b: 96 vs. 92% as upregulation, and does not follow dose-response curve). Further, Fig. S2c appears mislabeled as Fig. S3c.

Response: In the new version this was properly addressed. Thank you.

4) The labels for the flow cytometry in Fig. 3 should be made more legible.

Response: Done, thank you.

5) Fig. 4b-i, should have a ladder and the expected size of the product should be indicated. Also, are multiple HERV-V1/V2 integration sites located across the genome? Does CRISPR/Cas9 silencing account for all of them?

Response: Fig. 4b-i content was moved to the supplemental information (new **S4e**). Now it is a part of the description of the models employed, including design, location in the gene for targeting, the effect observable for the confirmation of the gene ablation, among further relevant information.

HERV-V₁/V₂ are appeared as single genes in the genome localized to the chromosome 19. There is a single integration of this endogenous retrovirus in the human genome. This aspect is in congruence with the ChIP-seq analysis.

6) Higher power insets in Fig. 7b would be helpful to visualize tumor cells.

Response: We provided additional pictures of the tumor histology, augmented.

7) Statistical analyses should be clearly described in all figure legends. Additionally, if statistical significance is claimed, error bars should be shown.

Response: Descriptive analysis was performed and reflected in all experiments. Thank you for this observation

8) The language used, at times, is somewhat colloquial (e.g. cells “commit suicide”), and there are translation errors in the text (e.g. mamma instead of breast). Please proofread.

Response: The language was corrected by our colleagues Marc Teipel and Crista Ochsenfarth, native English speakers.

We are really very grateful to this reviewer. His/Her recommendations and ideas improved our article significantly. Please, convey our thanks to him/her!

Answer to the Reviewer 2

We thank this reviewer for the very positive opinion on our work and the useful comments for improving the manuscript's quality.

1. "A major weakness is the study strongly relies on the SKOV3 cell line as a model to draw its conclusions. While some experiments (e.g. CRISPR/Cas9 knockout) may be difficult to perform in primary cells, other experiments, such as HERV-V1/V2 expression and downstream effects on signaling and apoptosis pathways, would be feasible in primary cells."

Response: This was fully addressed according to the recommendation of this reviewer. We studied the signaling of NFκB, Akt/PKB and Ras-MEK-ERK cascades in two cell lines and two primary ovarian carcinoma cells.

2. "In the results section "synergistic cytotoxic effects of HDACis and TLR7as in SKOV3^{WT} and primary ovarian cancer cells", the study mentions the use of three primary human ovarian cancer cell isolates (OvCa28, OvCa29 and OvCa236). However, data in Fig. 1 and Fig. S1 only show drug toxicity for two of these (OvCa28 and OvCa236), and isobolograms are only shown for OvCa236. In order to show the reproducibility of their findings, the authors have to present cytotoxicity and isobologram data for all three ovarian cancer isolates, and may comment on any observed differences between the three."

Response: Cytotoxic curves, Isobolograms in different combinations of HDACis/TLR7-8as were performed in two primary ovarian carcinoma cells isolated from ascites. We did not continue the study in OvCa29 since this cell line acquired slower growth patterns (no mycoplasma infections) and it was not possible to repeat the experiments at least 3x. In addition, we employed OvCar3^{WT}, which is a standard cell line commonly used in the ovarian carcinoma research. We did that for the accessibility of such studies by the community. Now the whole study was performed in two cell lines and two primary ovarian carcinoma cell models.

3. "In the results section "Simultaneous exposure to HDACis and TLR7as interferes with transcriptional expression of HERV-V1 but not of TLR7", the findings presented for **HERV-V1/V2 RNA** expression are key for most of the study's conclusions, but are only being shown for the SKOV3 cell line. In order to justify the conclusions made by the authors regarding the translational significance of the study, the HERV expression under drug treatment needs to be assessed in primary cell samples *in vitro*. This would ideally be done in the three ovarian cancer isolates (OvCa28, OvCa29 and OvCa236) already presented in Fig. 1 and Fig S1."

Response: We performed this experiment in OvCa236 cell line and observed similar effects (new **S3h**). We addressed this point inclusive in *in vivo* experiments. Since we investigated the antitumoral activity in other ovarian cell lines as for OvaCar3, we additionally searched for HERV-V₁/V₂ transcripts in the tumours remnants. For this purpose it was necessary to apply a low dosage of the drugs and combinations to obtain some tumoral mass for RNA isolation and

IHC studies. In addition we investigated at protein levels the diminution of HERV-V₁/V₂ protein in a kinetic over 72h and observed that the more incubation with the drug combination the more diminution in those viral proteins. By the way, these experiments were performed using 0.25% of the IC₅₀-values to obtain a sufficient cell density which allowed us protein isolation.

4. In the results sections “HDACi-TLR7a combination treatment is a potent, selective inducer of intrinsic apoptosis, Effects of HDACis and TLR7as on inflammatory factors and Combination of HDACis and TLR7as impairs signal transduction pathways”, these three paragraphs highlight potential mechanisms leading to the observed drug combination effects, namely induction of apoptosis and alterations in signaling pathways like NFκB, MEK/ERK and AKT. Alterations were shown to take place on the transcriptional (Fig. 5A) and translational/post-translational level (Fig. 5B, Fig. 6). However, the authors again only show data for the SKOV3 cell line, leading to the question whether any of the observed mechanisms could be recapitulated in other cell lines or primary samples. The authors need to address this question by repeating some of their mechanistic experiments in other cell models, preferentially primary cell isolates like the ones shown already in Fig.”

Response: We addressed this point as recommended. We analyzed the impact of the treatment on NFκB, Akt/PKB and Ras-MEK-ERK not only at protein level, but we investigated the phosphorylation steady state (post-translational modifications) of those transducers. We conducted the experiment in four models: SKOV3^{WT}, OvCar3^{WT}, OvCa236 and OvCa2810.

5. “In the results section “HERV-V1/V2 gene ablation with CRISPR/Cas9 significantly reduces apoptosis”, the presented data indicate successful gene editing with CRISPR/cas9 on the DNA level, as well as alterations in downstream effects (apoptosis, PARP cleavage), but do not confirm a change in HERV-V1/V2 RNA expression as a result of CRISPR/Cas9 treatment. HERVs are multi-copy loci and can have complex deletions, mutations and varying sequence homology. It can be expected that gene editing affects multiple loci, but leaves others intact, since editing efficiency is limited. Thus, confirmation of actual expression differences caused by CRISPR/Cas9 is essential. To justify their conclusions regarding the link between HERV transcript expression and downstream effects after drug treatment, the authors need to document alterations in HERV-V1/V2 RNA transcription in the CRISPR/Cas9-modified SKOV3 cells.”

Response: This recommendation was addressed. HERV-V₁/V₂ genes are present as single copy in the human genome (see results presented in new **Figure 5**). We investigated the levels of HERV-V₁/V₂ at RNA and protein levels. We also confirmed that those genes are apoptosis mediators, since once ablated, there is a significant decrease in apoptosis observed. This information can be now found in **Figure 4 c-e**.

Minor critiques:

1. “The drug concentrations throughout the manuscript are given in µg/ml, which makes it hard for readers to compare the presented data with other studies that use molar concentrations. The authors should – at least for their most important experiments – present drug concentrations in molar units (e.g. µM, nM) as well.”

Response: We reflected the dose-response curves as log of the concentration. The IC₅₀-values were depicted as molar concentrations. We apologize for giving that information in µg/ml

scale. We did this because it is more usual in the clinic and is more extrapolatable to animal models.

2. "Fig. 7B seems to be missing a description in the figure legend. This needs to be added, including an explanation of what the arrows in the presented images mean."

Response: We addressed this point. Thank you!

We want to thank this reviewer for his/her very positive criticisms and valuable comments on our manuscript.

Answer to the Reviewer 3

We appreciate the careful revision and the constructive suggestions provided by this reviewer and thank him/her for the inputs.

"1. The conclusions about NFκB are not supported by the results. Western analysis of NFκB subunits does not necessarily reflect the extent of NFκB activation. In order to make conclusions about NFκB activation, the NFκB activity should be analyzed by CHIP or a luciferase assay. In addition, it is not clear what NFκB subunit was analyzed; p65?"

Response: This topic was of paramount importance and we addressed the recommendations of this reviewer. We investigated this issue in different ways. The activity of NFκB was measured by luciferase assay (NanoLuc[®] purchased from Promega). Moreover, we investigated the expression of Rel A, Rel B, c-Rel and importantly NFκB1 and NFκB2 homodimers. In addition, we investigated the phosphorylation status of I-κBα which is hypophosphorylated under the action of HDACis/TLR7-8as. In the old version we presented in fact p65 (RelA). Now, new results can be found in **Figure 6**.

"2. HDI have been previously reported to induce expression of the pro-inflammatory chemokine IL-8 and other NFκB-p65 dependent genes in ovarian cancer cells by increasing the activity of IκB kinase. It would be interesting to determine whether the TLR7 agonists suppress the HDI-induced IL-8 expression in OC cells, since IL-8 has a tumor-promoting effect in OC."

Response: IL-8 is a very promiscuous cytokine upregulated in several cancer types, including ovarian cancers. As we addressed this point we learned that the combination reduce the levels of IL-8 transcripts in OvCar3^{WT}, primary ovarian carcinoma cell line OvCa236, but not always in SKOV3^{WT} cell line, which revealed very incongruent results compared to the other cells evaluated. A possible explanation could be the fact that IL-8 has different transcription factors distinct from NFκB1/RelA, like many cytokines also have. Nevertheless, taking into consideration that the cells die after this combination treatment, this effect could be less relevant in the whole analysis of the results. Another point to be taken into consideration is the analysis of IL-8 in *in vivo* systems, because this interleukin plays a pivotal role in the function of macrophages which in turn are necessary for the clearance of damaged cells as for example dying tumor cells.

“3. Materials and Methods: Western blotting and the source of the antibodies used should be described.”

Response: We addressed this point and included a description of Western blotting as well as the source of the antibodies employed in this investigation. We provided full information of all antibodies employed in this research in accordance with the journal policies.

Minor critiques:

“English should be corrected (e.g., First page, Financial support: “... grants awarded by ...”; Abstract, last line “...promising value for cancer treatments.”

Response: Done. The manuscript was proof read by our colleagues Marc Teipel and Crista Ochsenfarthn native English speakers.

Finally, we want to express our deep gratitude for the help provided by the reviewers. Their job has permitted us to improve the manuscript quality and present this investigation with high scientific level. Many thanks for the patience and help to all the people involved in the revision of this work.

Arrangement of figures in the main document of article entitled “*Enhanced Antitumoral Activity of TLR7 Agonists via Activation of Human Endogenous Retroviral Elements by HDAC Inhibitors*”.

Old figures	New figures
Figure 1: Consisted of three panels: Panel A: expression of HERVs in human ovarian cancer (expanded in new Figure 1) Panel B: IC₅₀ graphs (moved to new Figure 2) Panel C: Isobolograms (moved to new Figure 2)	Figure 1: a: contains expression of different HERVs in SKOV3 wild type and carboplatin resistant models in order to offer a scope on the differential expression of HERVs in relation to chemotherapy resistance. b: besides the expression of HERV-V₁/V₂ in ovarian carcinoma tumours and ovarian regular tissues, an isotype control was added
Figure 2: Consisted of three panels: Panel A: Effect of HDACis, TLR7/8as and their combination on transcription of HERV-V₁/V₂ (as recommended by reviewer, we replaced this figure for Figure S2a from old supplemental document and now is presented as Figure 5a (new), following the story). Panel B: Influence of HDACis, TLR7/8as and their combination on HERV-V₁/V₂ proteins. We expanded the results of this experiment and now the resulted FACS and Western blot analysis are moved to new supplementals S3f & g. Panel C: Effect of HDACis, TLR7/8as and their combination on transcription of TLR7. This figure is now new Figure 6b.	Figure 2: The original information of this picture comes in part, from old Figure 1, Panel B & C. We present all the dose response curves for romidepsin, vorinostat, imiquimod and vesatolimod as well as respective Isobolograms in SKOV3^{WT} cell line and OvCa236 primary cells. Complimentary information regarding OvCar3^{WT} and OvCa2810 regarding cytotoxic studies can be found in new S2a, c & d.
Figure 3: No major changes were made. But, we expanded those studies to other combinations that you can find in new S3 a & b. Panel A: Apoptosis pathway studies in SKOV3^{WT} after exposure to HDACis, TLR7/8as and their combination. Panel B: FACS studies. This panel was updated with better results and as per	Figure 3: We introduced the effect of romidepsin, vesatolimod and their combination on apoptosis pathway in primary cell line OvCa236 and moved effect of vorinostat, romidepsin and their combination data of SKOV3^{WT} to supplementals S3a & b. b: contains better ICC picture in which it is clear to observe the action of c-PARP localised to nucleus.

reviewers recommendations, isotype control was introduced.	c: updated version of old Figure 3, Panel B
Figure 4: Consisted of two panels. Panel A: was removed following the recommendation of reviewer 1. Panel B: T7 digestion (i) is moved to new supplementals S4e. All the detailed information regarding CRISPR/Cas9 ablation of HERV-V₁/V₂ can be found in new S4d-f. The information in Panel B ii-iv is now in new Figure 4.	Figure 4: It was upgraded with new results. Now, in this figure you can find the effects of the ablation of both TLR7 and HERV-V₁/V₂ separately on apoptosis to confirm the mediation of those elements. a: This picture reflects the reduced apoptosis induced by romidepsin and vesatolimod combination in TLR7 KO SKOV3 model as measured by FACS. b: reflects the same effect from a in WB c: reflects the reduced apoptosis induced by romidepsin and vesatolimod combination in HERV-V₁/V₂ KO SKOV3 model in Annexin assay as measured by FACS. This was the former Figure 4 Panel B ii in which we introduced error bars and distribution. d: This picture reflects the reduced apoptosis induced by romidepsin and vesatolimod combination in HERV-V₁/V₂ KO SKOV3 model as measured by FACS e: reflects the same effect from d in FACS
Figure 5: Consisted of two panels. Panel A: qPCR profile of inflammasome mediators. This can be found in new Figure 6c & e. Panel B: Study of combination treatment on NFκB and MyD88. This was moved and the new expanded results can be found in Figure 6a.	Figure 5: contains new results from ChIP-seq analysis a: HDACi dose dependent activation of HERV-V₁ and V₂. Following the recommendation of reviewer 1, we moved this figure from S2a (old) to this place to be consistent with the flow of the manuscript. b: New results from ChIP-seq reflecting information of HERV-loci and diffpeaks obtained after HDACi treatment. c: New figure. HERV-V2 localisation in chromosome 19 highlighting the structure of this gene.
Figure 6: Consists of two panels. The content of this figure is now found in Figure 7 which was expanded for other OC cell lines and primary cells isolated from ascites. Panel A: Influence of HDACis, TLR7/8as and their combination on Akt/PKB and RAS-MEK-	Figure 6: a: reflects the effects of romidepsin, vesatolimod and their combination in two ovarian carcinoma cell lines in MyD88-NFκB signaling. (expansion of old Figure 5 Panel B) b: The influence of HDACis, TLR7/8as and their combination on transcription of TLR7 as

ERK pathways. Panel B: Cartoon depicting the major events occurring after combination treatment. This was updated according to recommendations of reviewers and moved to new Figure 7b.	a relevant part of the analysed pathway. (moved from old Figure 2 Panel C). c: influence of HDACis, TLR7/8as and their combination on the gene expression of MyD88 adaptor protein and its interacting downstream elements. (part of old Figure 5 Panel A) d: New figure. NFkB activity as measured by luciferase assay. Experiments were performed in two cell lines as recommended by the reviewers. e: Gene expression of inflammatory factors after exposure to HDACis, TLR7/8as and their combination in SKOV3^{WT} cells. (part of old Figure 5 Panel A)
Figure 7: is now new Figure 8 in which new data was added	Figure 7: a: Influence of romidepsin, vesatolimod and their combination on Akt/PKB and RAS-MEK-ERK pathways in SKOV3^{WT} and OvCa236. The expansion of this pathway studies as recommended by reviewers in OvCar3^{WT} and OvCa2810 can be found in new S7a. b: Cartoon depicting the major events occurring after combination treatment. This was updated according to recommendations of reviewers. (old Figure 6 Panel B). We eliminated the elements which are not addressed in the manuscript. In addition, we incorporated the elements from MyD88-NFkB signaling according to new information gathered.
	Figure 8: originally Figure 7. In this figure, we incorporated qPCR and IHC studies on the expression of HERV-V₁/V₂ after treatments in OvCar3^{WT} xenograft model.

REVIEWERS' COMMENTS:

Reviewer #1 (Remarks to the Author):

The authors have satisfactorily addressed my concerns

Reviewer #2 (Remarks to the Author):

In their revised re-submitted manuscript, the authors have addressed points raised by the reviewer in great detail. A point-by-point rebuttal of all initial reviewer comments has been attached. The authors address comments concisely, and supplement additional experimental data, as well as explanation of their findings in the revised manuscript. They provide meaningful additions to the presented study, raising its overall quality. They also provide a point-by-point comparison of each manuscript figure, making it easier to track changes in figure layout in the re-submitted work.

Most notably, they have addressed criticism regarding the use of only one single cell line (SKOV3) in their experiments. The study in its current state has now been performed with two primary ovarian cancer cell isolates (OvCa236 and OvCa2810), and two ovarian cancer cell lines (SKOV3 and Ovar3), significantly corroborating conclusions from the in vitro assays shown in figures 1,2,3,6 and 7 as well as supplementary figures S2, S3, S5, S6 and S7. This provides broad confirmation for the experimental findings presented. In particular, the authors add further to their findings by including ChIP-seq assay (Fig. 5, S5), and additional Western Blots for signaling pathway analysis (Fig. 6, 7, S7). These additional data do not only support the authors' conclusions, but also add further layers of information for their findings by revealing chromatin modifications altered by HDACi treatment. As stated by the authors, primary ovarian cancer cell sample OvCa29 was excluded due to low growth rate of the cells. This is an understandable limitation, and exclusion of the cell sample is therefore fully justified.

New documentation on the HERV locus (Fig. 5c, S4) also clarifies some of the previous questions, and adds to the transparency of the manuscript. Especially from a genetics/genomics perspective, the findings are now well documented and will be easier to compare to other studies of such repeat elements. In this context, RNA expression changes of HERV upon HDACi treatment, as well as altered chromatin modifications at the HERV locus, have been highlighted (Fig. 5). This is a key finding of the study, and it is now presented and confirmed much more comprehensively.

Overall, the authors have addressed and clarified the points of criticism raised by the reviewer, especially the reproducibility of data in both cell lines and primary samples.

Reviewer #3 (Remarks to the Author):

The authors satisfactorily addressed most of the points raised in the previous review. This reviewer recommends acceptance of the manuscript.